# Composite Nanoarchitectonics of Photoactivated Titania-Based Materials with Anticancer Properties

**DOI:** 10.3390/pharmaceutics15010135

**Published:** 2022-12-30

**Authors:** Nefeli Papadopoulou-Fermeli, Nefeli Lagopati, Natassa Pippa, Elias Sakellis, Nikos Boukos, Vassilis G. Gorgoulis, Maria Gazouli, Evangelia A. Pavlatou

**Affiliations:** 1Laboratory of General Chemistry, School of Chemical Engineering, National Technical University of Athens, Zografou Campus, 15789 Zografou, Greece; 2Laboratory of Biology, Department of Basic Medical Sciences, Medical School, National and Kapodistrian University of Athens, 11527 Athens, Greece; 3Laboratory of Histology-Embryology, Molecular Carcinogenesis Group, Department of Basic Medical Sciences, Medical School, National and Kapodistrian University of Athens, 11527 Athens, Greece; 4Section of Pharmaceutical Technology, Department of Pharmacy, School of Health Sciences, National and Kapodistrian University of Athens, 15771 Athens, Greece; 5Institute of Nanoscience and Nanotechnology, National Center for Scientific Research “Demokritos”, 15310 Agia Paraskevi, Greece; 6Biomedical Research Foundation, Academy of Athens, 11527 Athens, Greece; 7Clinical Molecular Pathology, Medical School, University of Dundee, Dundee DD1 9SY, UK; 8Molecular and Clinical Cancer Sciences, Manchester Cancer Research Centre, Manchester Academic Health Sciences Centre, University of Manchester, Manchester M20 4GJ, UK; 9Center for New Biotechnologies and Precision Medicine, Medical School, National and Kapodistrian University of Athens, 11527 Athens, Greece; 10Faculty of Health and Medical Sciences, University of Surrey, Guildford GU2 7YH, UK; 11School of Science and Technology, Hellenic Open University, 26335 Patra, Greece

**Keywords:** composite materials, Ag-TiO_2_, sol-gel, visible light, photocatalysis, PNIPAM, microgel, anticancer activity, nanoarchitectonics

## Abstract

The synthesis of titania-based composite materials with anticancer potential under visible-light irradiation is the aim of this study. In specific, titanium dioxide (TiO_2_) nanoparticles (NPs) chemically modified with silver were embedded in a stimuli-responsive microgel (a crosslinked interpenetrating network (IP) network that was synthesized by poly (N-Isopropylacrylamide) and linear chains of polyacrylic acid sodium salt, forming composite particles. The ultimate goal of this research, and for our future plans, is to develop a drug-delivery system that uses optical fibers that could efficiently photoactivate NPs, targeting cancer cells. The produced Ag-TiO_2_ NPs, the microgel and the composite materials were characterized through X-ray diffraction (XRD), Fourier transform infrared spectroscopy (FT-IR), micro-Raman spectroscopy, ultraviolet-visible spectroscopy (UV-Vis), dynamic light scattering (DLS) and transmission electron microscopy (TEM). Our results indicated that Ag-TiO_2_ NPs were successfully embedded within the thermoresponsive microgel. Either Ag-TiO_2_ NPs or the composite materials exhibited high photocatalytic degradation efficiency on the pollutant rhodamine B and significant anticancer potential under visible-light irradiation.

## 1. Introduction

The bioactivity of photoinduced titanium dioxide (TiO_2_) has been studied in recent years in order to propose an alternative therapeutic approach, minimizing the side effects caused by conventional cancer treatments [1]. Pure TiO_2_ can induce cyto- and genotoxicity through programmed cell death, eradicating cancer cells thanks to its photocatalytic properties [2,3]. TiO_2_ is widely used in photocatalysis because it is considered the most suitable photocatalyst thanks to its high photocatalytic efficiency, physicochemical stability, nontoxicity and chemical inertness [4]. The wide energy band gap (E_g_) of TiO_2_ (E_g_ = 3.2 eV) allows photon absorbance, mainly in the UV (ultraviolet) range of the spectrum of electromagnetic radiation, resulting in the electron-hole-pair production that participates in redox reactions, degrading organic species [5,6,7,8,9]. It is well known that Ag NPs are widely used in medical applications thanks to their antibacterial activity [10]. Thus, TiO_2_ and Ag together led researchers to focus on the development of silver-doped titania biomedical devices, coated surfaces for food preparation, filters for air conditioning and other applications [11].

Silver is able to trap the excited electrons from TiO_2_, leaving the holes available for the degradation of organic species [12,13]. Silver also permits the extension of TiO_2_ responses, including the visible region of the electromagnetic radiation spectrum [14]. Composite materials that consist of semiconductors and noble metals are considered surface plasmon resonance (SPR) active under visible light, thanks to the presence of the noble metal [15]. Photoactivated Ag-TiO_2_ NPs with visible-light exhibit catalytic redox reactions; thus, they are widely known as plasmonic photocatalysts [9,13]. Furthermore, the need for the chemical modification of TiO_2_ by noble metals was based on using a wider range of solar radiation, including visible light [5,9,13,16].

Plasmonic photocatalysts have attracted significant scientific attention in order to be used in the catalytic degradation of several organic pollutants or dyes through visible-light photocatalysis [13,17]. In order to recycle the catalysts and reduce the toxicity that is related to the nano-size of the particles, the point is to immobilize the selected photocatalyst on various substrates, such as silica and polymeric materials [18,19]. An ideal photocatalytic system must possess the following characteristics [20]: (a) it should facilitate strong interactions between the supporting material and the photocatalyst, preventing them from leaching during various experimental processes; (b) independent of the technique that is selected in anchoring the photocatalyst on the support, it is important for a photocatalyst to be reactive, providing a large surface area; (c) the resulting photocatalyst on a support should also be stable over a given time period; and (d) the support must be resistant to the degradation caused by the produced reactive oxygen species during the photocatalysis process.

A specific category, that of stimuli-responsive sensitive polymers, especially with a gel structure, offers a cost-effective alternative to conventional processes for medical and industrial applications [21,22,23,24]. Stimuli-responsive polymers are very sensitive even to slight changes that are observed in an environmental condition, such as temperature, salt concentration or pH, showing a sharp change in their properties and behavior [25]. This different performance can be used in the preparation of so-called smart gel (microgels), which are crosslinked polymeric particles. These polymeric materials can be considered as hydrogels if they consist of water-soluble/swellable polymer chains [26].

Moreover, pH-responsive polymeric materials that are composed of acrylic acid derivatives were introduced by M. Palasis [27] because they can attract negatively charged therapeutic agents. When these materials are found at pH values that are above their pKa, then they become mainly uncharged and thus can controllably release the embedded therapeutic factors [28]. Dual-sensitive microgels are sensitive to both temperature and pH and can be prepared through the combination of a polyelectrolyte comonomer and a thermosensitive monomer, such as NIPAM. Additionally, multiresponsive microgels can be developed with the proper combinations of the monopolymers that are used [29], and these materials that can simultaneously respond to more than one stimulus are very promising in biomedical applications [30]. Using polymers that are responsive to various stimuli and creating efficient drug-delivery systems contribute to the increased therapeutic effect. Thus, dual-stimuli-responsive polymers are suitable for theragnostic approaches (a combination of diagnostics and therapeutics) [31].

Among several synthetic thermosensitive hydrogels, the PNIPAM (poly (N-isopropylacrylamide)) hydrogel is widely studied with a lower critical solution temperature (LCST) of ~32 °C [32,33,34]. Below LCST, PNIPAM hydrogel is able to absorb a high amount of water in a transparent swollen state. Because the temperature increases to temperatures above LCST, PNIPAM hydrogel would transition to the collapsed volume state following a drastic, discontinuous process [35,36,37,38,39]. To prepare aqueous microgel PNIPAM particles with ideal properties, the precipitation polymerization method is commonly applied [40]. This method allows the development of PNIPAM microgel particles with a controlled size, a narrow hydrodynamic diameter distribution and enhanced colloidal stability [41]. Postpolymerization modification reactions can achieve the encapsulation of small organic molecules, inorganic NPs, biopolymers or synthetic polymers inside the microgel network [42].

Previous studies conducted by our research group have shown that TiO_2_ [43,44], N/TiO_2_ [3] and Ag/TiO_2_ NPs [45] may occasionally induce apoptosis on highly metastatic breast cancer cells, after photoactivation with UV-A light. Additionally, we have already shown that among the common crystal forms of TiO_2_, which are anatase, rutile and brookite, pure anatase (100%) was more bioactive, inducing apoptotic cell death on breast cancer cells, than a combination of anatase and rutile (75% anatase/25% rutile) [43]. Recently, we tried to estimate the biological effect of polymerically embedded TiO_2_ NPs under visible-light irradiation, resulting in the development of efficient pNipam-co-PAA/Ag-TiO_2_ composite materials (using nitrogen and iron as dopants), acting as a thermoresponsive drug platform and operating under similar conditions to those that are found in the human body. We noticed a significant cytotoxic effect on highly metastatic breast cancer cells, which are associated with oxidative stress [32]. According to our previous systematic and very promising results, we chose a different combination of the well-studied materials that we used, in order to optimize the properties of the produced composite. Thus, focusing on the present study, we aimed for the development of an innovative drug-delivery system that was designed in order to possess anticancer activity under visible light. Therefore, thermoresponsive Ag-TiO_2_-based composite materials were prepared according to the sol-gel method. The ultimate goal of this research, and for our future plans, was to develop a system using optical fibers that could efficiently photoactivate nanoparticles—targeting cancer cells, avoiding the undesirable effect of the conventional therapeutic approaches and reducing the cost for the health system, thereby supporting those treatments. Hence, in order to embed the Ag-TiO_2_ NPs, we synthesized a stimuli-responsive polymer microgel by using a NIPAM monomer with interpenetrating linear chains of polyacrylic acid sodium salt to stabilize the Ag-TiO_2_ NPs. Interpenetrating polymer networks (IPNs) are a category of polymer blends that are formed as a network, provided that one of the components is polymerized and/or crosslinked in the presence of the other [22,27,34,38]. IPNs are composed of multiple polymers in a network form. They are bound by permanent entanglements, and only a few accidentally formed covalent bonds between the chains of the two polymers [32]. Techniques such as X-ray diffraction (XRD), Fourier transform infrared (FT-IR), micro-Raman, ultraviolet-visible (UV-Vis) spectroscopy, dynamic light scattering (DLS) and transmission electron microscopy (TEM) were applied in order to confirm that the produced NPs and composite materials have the proper physicochemical characteristics and morphology. Furthermore, the degradation of the rhodamine B (RhB) pollutant was evaluated for the produced nanopowder and the composite materials by using visible-light irradiation. For the analysis of their anticancer behavior, two breast cancer epithelial cell lines (MCF-7 and MDA-MB-231) and normal human embryonic kidney cells (HEK 293) were cultured and treated with the produced materials by using irradiation with visible light, and cell proliferation and cytotoxicity assays were employed.

## 2. Materials and Methods

### 2.1. Preparation and Synthesis

#### 2.1.1. Microgel

The PNIPAM/PAA microgel was formed by surfactant-free precipitation polymerization. A monomer of NIPAM (9 × 10^−2^ mol, N-Isopropylacrylamide > 98%, TCI, Chennai, India) was adequately purified and then recrystallized using hexane, polyacrylic acid sodium salt (1.6 × 10^−2^ mol, MW 15,000 g/mol, Sigma-Aldrich, Darmstadt, Germany) and MBA (2.6 × 10^−4^ mol N,N-Methylenebisacrylamide > 99%, Alfa Aesar, Ward Hill, MA, USA), which was dissolved in an aqueous solution (final volume at 200 mL). Acting as a cross-linker for this synthesis, MBA was selected. The reaction mixture was then purged for 1 h with pure N_2_ (99.99999%) and afterward heated to 70 °C. After this step, KPS (7 × 10^−5^ mol potassium persulfate > 99%, Fisher Chemicals, Hampton, NY, USA) was also added and the polymerization process, starting at a duration of 6 h, under vigorous stirring (Figure 1). The size behavior of the microgels was estimated by dynamic light scattering (DLS) at 173° optics (Zeta Sizer nano S, Malvern Inst., Malvern, UK).

#### 2.1.2. Inorganic Ag-TiO_2_ NPs

Employing the bottom-up sol-gel procedure, the synthesis of the TiO_2_ nanoparticles, chemically modified with silver, was realized. Thus, 25 mL of titanium isopropoxide (97%, Sigma-Aldrich, Darmstadt, Germany) was added to acetic acid (48 mL) (99–100%, Chem-Lab NV, Zedelgem, Belgium) under stirring. Deionized water (150 mL), including 5 mol of 99.8% silver nitrate (Fisher Scientific Chemical, Loughborough, UK), was then added to the solution, drop-wise under vigorous stirring. After 5 h of stirring, the solution was a clear sol (titanium isopropoxide, acetic acid and water were used according to 1:10:100 molar ratio). The new solution was then heated until the solvent had been completely evaporated. Afterward, the prepared gel was consequently calcinated at 600 °C for 2 h. The resulting nanopowder was triturated, then purified via rinsing and centrifuged in order to remove any remaining impurities. In the end, it had a gray color (Figure 2).

#### 2.1.3. Composite Nanoparticles

To form the encapsulated NPs, a suspension of chemically modified Ag-TiO_2_ na-nopowder was prepared. Specifically, 50 mg of Ag-TiO_2_ NPs was suspended in 20 mL of deionized water. The pH value of this suspension was adjusted to 3, using an HCL (1 Μ) solution to positively charge them. The first solution was added to 20 mL of microgel solution. After that, the pH was adjusted to ~6 by using a NaOH (1 Μ) solution, and the composite materials were collapsed. The composites were then cleaned by washing them three times and finally dried at 70 °C (Figure 3).

#### 2.1.4. Rhodamine B Solution

Rhodamine B (RhB) of an analytical reagent grade was purchased from Penta chemicals (Prague, Czech Republic); hence, there was no need for further purification. In addition, 1 L of RhB solution was prepared by using deionized water.

### 2.2. Characterization Techniques

The physicochemical properties of the produced materials and their structure was studied through various characterization techniques, particularly XRD, infrared spectroscopy and micro-Raman spectroscopy. X-ray diffraction (XRD) experiments were realized by using an angle 2-theta configuration, at a scan rate of 0.1°/min, in an ancle range of 20°–80°, applying Cu-Kα radiation (30 kV, 15 mA) (λ = 1.5418 Å) (XRD by Bruker D8 Advance, Yokohama, Japan). To perform Fourier transform infrared spectroscopy (FT-IR), a Brucker alpha II, platinum ATR was used, and 16 scans were performed to obtain each spectrum. The resolution was 4 cm^−1^. An ultraviolet-visible (UV-Vis) spectrometer (Jasco UV/Vis/NIR Model name V-770, Tokyo, Japan), equipped with an integrating sphere, allowing diffuse reflectance measurements, was used to measure the energy band gap of the produced particles. The InVia model from Renishaw was the micro-Raman device that was used (Renishaw, InVia, Wotton-under-Edge, Gloucestershire, UK). A solid-state laser (λ = 532 nm) as well as a high-power near-infrared (NIR) diode laser (λ = 785 nm) were utilized as excitation sources. The measurements were acquired at room temperature (RT) and in backscattering configuration. In order to avoid heating the samples, low excitation power was applied. Additionally, the beam of the laser was directed, and in order to be focused on the samples, an 50-times short-distance magnification lens was used. An internal SI reference was used for the calibration of the frequency shifts. For each sample that was exposed for 10 s, a number of 2–3 spots were acquired, with 2–10 accumulations.

The hydrodynamic diameter of the produced composite particles and the corresponding zeta potential of particles suspended in deionized water zeta potential were evaluated through dynamic light scattering (DLS) using a Zeta Sizer nano S (Zeta Sizer nano S, Malvern Inst., Malvern, UK). Finally, the morphology of the composite materials was studied via TEM (TEM-FEI Talos F200i field-emission (scanning) ThermoFisher Scientific Inc., Waltham, MA, USA), which was operating at 200 kV. The device was equipped with a windowless energy-dispersive spectroscopy microanalyzer (6T/100 Bruker, Hamburg, Germany) [32].

### 2.3. Photocatalytic Test

Evaluating the photocatalytic efficiency of the Ag-TiO_2_ NPs and the composite materials by using illumination with visible light, a series of experiments was conducted. Visible irradiation was emitted by a system of four parallel daylight lamps (15 W, Sylvania, Wilmington, NC, USA) (3 mW·cm^−2^, 350–750 nm) that were placed in a lab-made photoreactor. The samples were placed in glass containers, in a stand at distance of 10 cm from the lamps. RhB was the pollutant that was studied, testing its photocatalytic degradation when irradiated with visible light. A standard procedure was applied before each photocatalytic run, including the saturation of the solution in oxygen, by bubbling O_2_ gas through it for 2 h and maintenance in the dark for 24 h, in order to achieve an equilibrium between adsorption and desorption. Then, 5 mg of the Ag-TiO_2_ NPs or the composites were placed in the appropriate glass containers with the addition of 5 mL of RhB solution (0.01 M). All this experimental procedure was held under continuous stirring. During the photocatalytic test that lasted for 150 min, every 30 min, an amount of RhB was taken in order to measure its concentration. The ratio of *C*/*C*_0_ pollutant concentrations was obtained by the evaluation of the ratio of the absorption that was measured at each time point (A) to the initial (A_initial_) absorption [32,46]. A statistical analysis was conducted by applying the nonparametric Kruskal–Wallis statistical analysis; *p* < 0.05 was considered statistically significant.

### 2.4. Biological Anticancer Effect

#### 2.4.1. Cell Cultures

Three cell lines, HEK293 (ATCC, LGC Standards GmbH, Wesel, Germany (ACRL-1573), human epithelial kidney embryonic cells, normal), MDA-MB-231 (ATCC, HTB26), highly invasive, human epithelial breast adenocarcinoma) and MCF-7 (ATCC, HTB-22), Michigan Cancer Foundation (MCF)-7 (epithelial breast adenocarcinoma, low metastatic), were cultured in cell-culture dishes, using the appropriate media (Dulbecco’s modified Eagle’s medium (DMEM) (Gibco BRL, Life Technologies, ThermoScientific, Paisley, UK)), which were supplemented with 10% fetal bovine serum (FBS), 1% L-glutamine, 1% sodium pyruvate and antibiotics (Gibco, Paisley, UK). The cell cultures were maintained through incubation at 37 °C, 99% humidity, in a 5% CO_2_ incubator. For the trypsinization of cells, a mixture of trypsin–EDTA at 0.05%/0.02% (*w*/*v*) (Gibco, UK) was used [45].

#### 2.4.2. Estimation of Cell Proliferation Rate

The effect of all the produced materials on the proliferation of the cell population of the cultured cells was studied. Hence, for the needs of cell counting, ~100,000 cells/well were needed to be cultured in 6-well plates and incubated for 24 h. The day after plating, increasing concentrations of Ag/TiO_2_ NPs, PNIPAM/PAA salt and PNIPAM/PAA microgel/Ag-TiO_2_ were added. Then, the samples were photoactivated with visible light for 30 min using an incorporating venting system to avoid any thermal effects. It was crucial to select the minimal time of illumination of the NPs and the composite materials, as well as the optimal concentrations. Hence, the experiment needed to be repeated several times, under different conditions. When, finally, the ideal conditions were chosen, the experiment was conducted three times in triplicate. Staining with Trypan Blue and counting via a hemocytometer (Neubauer, Corning, the Netherlands) through optical observation (OLYMPUS IM, Olympus Deutschland GmbH, Hamburg, Germany) was performed every 24 h, to create growth-rate graphs [47]. A statistical analysis was implemented employing the nonparametric Kruskal–Wallis method; *p* < 0.05 was considered statistically significant [48].

#### 2.4.3. Cytotoxicity Test

The percentage of the viable cells was estimated through MTT colorimetric assay [3-(4,5-dimethylthiazol-2-yl)-2,5-diphenyl-tetrazolium bromide] (Thiazolyl Blue Tetrazolium Bromide M5655, Sigma-Aldrich, Darmstadt, Germany), as described previously [43,44]. Cells treated with 0.75 mg/mL of *cis*platin were considered as the positive control of the series of experiments. A blind sample including a cell-culture medium without cells, untreated cells (negative control) and the cell samples was irradiated with visible light, without Ag-TiO_2_ NPs or composites (extra negative control). Under the same conditions, the experiment was finally repeated three times in triplicate. A statistical analysis was performed using the nonparametric Kruskal–Wallis test; *p* < 0.05 was considered statistically significant [48].

## 3. Results and Discussion

### 3.1. Characterization of the Nanoparticles and the Composite Materials

#### 3.1.1. XRD Analysis

XRD was principally used to investigate the Ag-TiO_2_ NPs crystallinity but also the crystallinity of the produced microgel powder and the composite material. Figure 4 shows that anatase is the dominant crystal phase of Ag-TiO_2_. In specific, the main and highest diffraction peak of anatase was detected at 2θ = 25.35°, which belongs to (101) crystal plain. The other detected peaks of anatase phase were found in accordance with the PDF No 03-065-5714 [6,7,12,16,33,49]. According to the literature, the peaks of Ag-TiO_2_ NPs at 2θ, specifically 25.3°, 37.8°, 48.0°, 53.9°, 55.1°, 62.7°, 68.8°, 70.2° and 75.0°, seemed to correspond to the (101), (004), (200), (105), (211), (204), (116), (220), (213) and (215) crystal plains of anatase [50]. These intense peaks reflected the high crystallinity of Ag-TiO_2_/microgel and confirmed the successful loading of TiO_2_ into the PNIPAM microgel polymeric matrix, decreasing the crystal distances [51]. There is a characteristic peak at 2θ = 46.2° that corresponds to the formation of silver phase, according to the PDF-2-2003 [52,53].

No diffraction peaks were observed that correspond to the existence of impurities, such as Ag or Ag oxides, confirming that chemical modification did not disturb the anatase phase. Moreover, the fact that there is an absence of any impurity phases indicated that the integration of the silver ion was successful in the matrix of the TiO_2_ [54].

Owing to the low concentration of Ag and overlapping by the TiO_2_ characteristic peak (2θ = 37.8°) at 2θ = 38.1°, the corresponding peaks appeared relatively weak in the XRD pattern. The major diffraction peaks shifted toward the lower 2θ values that are obtained and became broader as the doping ion concentration increases. This phenomenon can be attributed to the lattice strain that is present in the samples [50,54] and attributed to the embedment in the semicrystal polymer network. With regard to the PNIPAM/PAA salt, the XRD patter shows a single broad diffraction peak centered at 2θ of ~21°, which is characteristic of amorphous PNIPAM [51,55]. In general, the crystallinity of pure PNIPAM is 0%, so the XRD pattern indicates a totally amorphous situation. In contrast with pure PNIPAM, the grafting-PNIPAM polymer microgel indicates an increased crystallinity. This suggests the formation of crystalline regions, as a result of grafting process, which were present as a consequence of the existence of H-bonding interactions between the OH groups of pectin chains and the NH groups of PNIPAM [56]. So hybrid microgels can be crystallized, but pure microgels do not exhibit any peak in XRD spectra [37]. The broadness in peak is due to copolymerization [57] and confirms the grafting of poly(acrylic acid) sodium salt chains onto PNIPAM. Additionally, the pattern exhibited a sharp crystalline peak at 2θ = 26.7°, 27.8°, ~30°, 39.8° and other microcrystalline peaks. Thus, the pattern suggests that the graft-polymerization process and the cross-linking created crystalline given that the distance between the molecules was reduced [57,58]. Possible slight lattice deformation was caused by the bonding of Ag-TiO_2_ and the polymer [59]. However, the diffraction peaks were broad, indicating that the crystal size was very small. The mean crystal size of the produced Ag-TiO_2_ NPs was estimated by Scherrer’s equation [32]. Thus, through these measurements and calculations, the average crystallite size was estimated at 9.52 nm.

#### 3.1.2. FT-IR Analysis

The characteristic of the PNIPAM gels is the three regions of wavenumbers: 1680–1500, 1420–1350 and 1300–1100 cm^−1^ [60]. The origin of the peaks and the differences between the spectra appear as consequences of the water exchange and the spontaneous reswelling transition of PNIPAM microgels and the ionization. The peak at 1713 cm^−1^ indicates the coexistence of hydrogen-bonded (HB) and non-hydrogen-bonded (NHB) carboxyl groups. The intensity of this peak at 1713 cm^−1^ is not large enough but exists. As Figure 5 depicts, four peaks (1745, 1731, 1715 and 1695 cm^−1^) are detected. The peak that corresponds to the C=O stretching vibration between the NHB carboxyl group is present at 1740 cm^−1^ [60,61]. The peak of the C=O stretching vibration of the HB group (dimeric form) is observed at a lower wavenumber (1710 cm^−1^) [60,61].

The peaks that appeared at 1745 and 1731 cm^−1^ may be associated with the C=O NHB carboxyl group’s stretching vibration (“free” C=O). The peaks at the region of 1715 and 1695 cm^−1^ could be assigned to the C=O stretching vibration of the HB carboxyl groups (C=O, H-O). A sharp peak at 1644 cm^−1^ is observed in the spectrum that corresponds to the amidic C=O NIPAM groups. The coupled peaks that correspond to the C=O stretching vibration and the O-H bending vibration in the hydrogen bonding between two carboxyl groups of the monomer lead to the formation of the broad peak at 1300–1200 cm^−1^. For the major bands of amide I and amide II (1680–1500 cm^−1^) [61,62], the one corresponding to amide I (located between 1680 and 1580 cm^−1^) is related to the C=O stretching vibration of the amide group of NIPAM. The amide II band (located between 1580 and 1500 cm^−1^) seems to be derived from the N-H bending vibration of the amide group of NIPAM. For the region of 2800–3800 cm^−1^, the width and the corresponding intensity of the bands is related to the absorbance of water, so the broad absorptions around 2800–3400 cm^−1^ are relevant to the O-H bonds [33,35]. The embedded Ag-TiO_2_ NPs reduce the capacity of the network in water absorption, possibly because the NPs have entered the pores of the network. The reduction in the intensity in the 2500–340 cm ^−1^ bands indicates the reduction in the water’s absorbance. The peaks that appeared in the region of 400–900 cm^−1^ are characteristic of the existence of a O-Ti-O lattice and a Ti-O stretching mode peak of around ~1630 cm^−1^ and correspond to the bending vibration of the H-O bond, which confirms that there is a formation of a metal oxygen bond [54,63,64].

#### 3.1.3. Raman Analysis

Figure 6a demonstrates several characteristic bands at 142, 196, 395, 515 and 637 cm^−1^, corresponding to E_g(1)_, E_g(2)_ B1_g(1)_, A1_g_ and E_g(3)_, submodes of the anatase crystal phase of commercial Evonik p25 TiO_2_, respectively. Regarding the Ag-TiO_2_ NPs, it seems that the spectrum is characterized by Raman peaks similar to those of pure TiO_2_, with minor differences. Probably, for these red shifts, Ag NPs modification is responsible, which tends to construct the Schottky barrier [65]. The peak intensities were found to be decreased, whereas the width of the peak increased, thanks to the lattice distortion and the presence of defect levels [66]. Additionally, the Raman spectra of Ag-TiO_2_ NPs exhibit no other peaks related to the brookite/rutile phase, confirming that all particles are in a single anatase phase [12,32]. The main Ag-O symmetric stretching (SS) vibration band of Ag_2_O expected at 490 cm^−1^ might be masked under the 515 cm^−1^ strong anatase band. Furthermore, it seems that Ag-TiO_2_ NPs succeed at preserving the anatase structure. This suggestion means that Ag dopants were incorporated into the structure of the TiO_2_ framework. Raman bands at 146 cm^−1^ slightly shifts toward increasing wavenumbers, increasing the silver ion. Generally, the shift that is observed in a Raman peak appears to be due to the detectable alternation in the size of the particles, their structure and the nature of defects. However, the broadening of the peaks is also depicted. This could be attributed to the particle size effect on the force constants and also to the vibrational amplitudes [54].

Changes were detected on the bands of PNIPAM, which are sensitive to hydrogen bond variations and can be explained through the different interactions between polymers, as well as among the polymer side groups and the available water molecules close to them. Thus, the Raman spectrum is characterized by an SS of CH_3_ (2880 cm^−1^), an SS and an antisymmetric stretching (AS) of CH_2_ (2920 cm^−1^ and 2945 cm^−1^, respectively) and an AS of CH_3_ (2988 cm^−1^) (see Figure 6b). In general, a significant red shift of the spectral weight between the two main peaks was related to the CH_2_ SS and AS modes of the methylene group. The intensity ratio between the SS and AS modes (lower and higher frequency, respectively) corresponds to the density of the lateral packing of any polymer chain [32,67,68]. The signals at 1650 cm^−1^ correspond to the carbonyl group and appear in both monomers. Additionally, the detected signal at 2920 cm^−1^ that corresponds to the Raman red shift of the methyl group is detected only in the Raman spectrum of the NIPAM monomer [32,61,69].

As we indicated in our previous work, in the frequency region of 2850–3050 cm^−1^, the detected peaks were assigned to the existence of different stretching modes, namely C-H of the molecule NIPAM in the hydrate state, at 2880 cm^−1^ to the SS of CH_3_, whereas peaks at 2920 cm^−1^ and 2945 cm^−1^ correspond to the SS and AS of CH_2_, respectively. Finally, the obtained peak at 2988 cm^−1^ is related to the AS of CH_3_ [32].

#### 3.1.4. Energy Band Gap Estimation

The energy band gap of a semiconductor indirectly determines the energy that should be provided to excite an electron that leaves the valence band (VB) and transmits to the conduction band (CB). An accurate evaluation of the energy band gap is crucial to predicting the photophysical and photochemical properties of semiconductors [70]. The optical properties were analyzed in the wavelength range of 300–850 nm by UV-Vis diffuse reflectance spectra (DRS). The powder reflectance and the E_g_ were measured through the Kubelka-Munk (K-M) method, as it was previously described [32] (Figure 7a,b).

The E_g_ of the commercial pure Evonik P25 was measured, which was E_g_ = 3.1 eV [32], and the produced Ag-TiO_2_ NPs revealed that there is a decrease in E_g_ (E_g_ = 2.32 eV), in comparison with pure anatase, owing to chemical modification with Ag. This finding is perhaps associated with enhanced photocatalytic behavior visible-light irradiation, because under the chemically modified photocatalyst needed lower energy for its photoactivation, overcoming the E_g_ [70].

#### 3.1.5. Dynamic Light Scattering (DLS)

The determination of the hydrodynamic diameter (D_h_) of the polymeric IP network (microgel PNIPAM/PAA salt) particles was achieved through DLS, within a temperature range of 25–45 °C, at a constant pH value of ~7.4.

A red laser beam that was operating at 633 nm was used for the measurements. An amount of 1 mL of the microgel PNIPAM/PAA salt was placed in a proper DTS1070-type capillary cell (Malvern Instr., Malvern, UK) and measured. Volume phase transition temperature (VPTT) was defined as a direct, perpetual transition in the D_h_ volume, at the swollen phase and the collapsed phase [26,32,38]. The DLS data that are shown in Figure 8 indicate that the collapsed D_h_ of the microgel was ~322 nm, whereas the swollen D_h_ was ~720 nm. The VPTT was evaluated by plotting the first derivative of the particle diameter versus temperature [32]. We estimated the VPTT of the sample to be around 37.6 °C, and thus, a shift toward higher values was observed, compared with the VPTT that was reported for PNIPAM. Hence, this VPTT is considered suitable for applications in drug-delivery systems in biological environments, such as the human body. The selection of the copolymer and the optimal selection of the time interval among the addition of the two monomers of the graft-polymerization process can result in this shift. In temperatures below the VPTT, in an aqueous solution, the chains are still soluble, thanks to the presence of hydrogen bonds that are formed among water and amine chains. In temperatures over the VPTT, a significant amount of water is finally discarded from the microgel; thus, the D_h_ volume appears to shrink [32].

The aqueous solution of Ag-TiO_2_ NPs was measured at a pH of 7.4 and the corresponding zeta potential measurements revealed very stable suspensions, as demonstrated in Figure 9 (ZP = (−53.5 ± 6) mV).

#### 3.1.6. TEM Analysis

The observation of the morphology of all the produced composite NPs and the IP network microgel/Ag-TiO_2_ NPs was performed with transmission electron microscopy (TEM). The images of the PNIPAM/PAA microgel/Ag-TiO_2_ are shown in Figure 10a–e. PNIPAM/PAA microgel/Ag-TiO_2_ composite was examined using TEM to visually determine the extent of Ag-TiO_2_ loading [38]. In the sample, several microgel particles appeared as clear objects, with sizes of approximately ~600 nm, but some particles baffled each other because of the TEM measurement, which led to a concentrated sample of hybrid microgels dispersion, under the intensity of the electron beam [71]. The measured diameter of the hybrid microgel particles might be lower than that was measured by DLS (see Figure 8), which might be attributed to a significant decrease in D_h_ on the drying microgel sample, placed on the grid for TEM measurement [71]. The observed dark black spots in the TEM image represent Ag-TiO_2_ nanoparticles that were loaded into the microgel (Figure 10a,c–e). Figure 10a,c,d show the typical morphology and surface of the produced composite Ag-TiO_2_/microgel NPs, compared with pure TiO_2_ (Figure 10f) and Ag NPs (Figure 10g). The TEM images of the hybrid microgels showed that Ag-TiO_2_ nanoparticles are uniformly distributed in the PNIPAM/PAA salt and not aggregated, as is clear in Figure 10a,b. Regarding the Ag-TiO_2_ NPs, the TEM images (Figure 10a,c–e) clearly show the sphere-like synthesized material. Observation revealed that TiO_2_ NPs were crystallized well, with a lattice-fringe spacing of 3.55, 2.34, 1.70, 0.24, 1.50 and 1.89 nm, and these findings well match the d-spacing of (101) (112) (105) (004) (213) and (200) planes of anatase TiO_2_. Ag NPs could be found on the surface TiO_2_ (in a decorated type) (Figure 10d,e,h,i) [11]. Ag NPs appear with a lattice-fringe spacing of 0.2 nm, which adequately matches the d-spacing of (002). The dopant Ag was well distributed. The Ag-TiO_2_ has many black–gray colored dots, in contrast. It is important to highlight that these TEM findings might verify the size scale of the Ag-TiO_2_ NPs, as was previously recorded by XRD [12,50,72]. The energy-dispersive spectroscopy (EDS) analysis of Ag-TiO_2_/microgel composite is also displayed in Figure 11. It seems that the Ag and Ti elements were incorporated and distributed in the polymer microgels, confirming that Ag-TiO_2_ NPs were successfully formed. Figure 10c,d clearly indicate the successful embedment of the Ag-TiO_2_ NPs in the polymeric network.

### 3.2. Photocatalytic Activity Experiments

#### 3.2.1. Photocatalytic Efficiency and Kinetics

RhB (C_28_H_31_CIN_2_O_3_) was chosen as a stable pollutant for the photocatalysis experiments to test the prepared TiO_2_ basis materials. RhB is a basic red dye of the xanthene class, which has been commonly used as a colorant in foodstuffs and in textiles. It is one of most hazardous types of textile waste and is considered as carcinogenic because it consists of radical compounds that potentially cause damage to the ecosystem [73]. Several inorganic semiconductor materials were used as photocatalysts to degrade wastewater, and many of them have been widely studied [8,9,13,74]. The photocatalytic experiments were implemented at room temperature and at pH of 7.4 (simulating the biological environment), in order to certify the photocatalytic ability of the Ag-TiO_2_ NPs and composites, before testing their biological effect on cancer cells. The photocatalytic efficiency of the produced materials was estimated under appropriate visible-light illumination.

Figure 12a depicts the photocatalytic activity of all the produced materials under visible irradiation for 150 min. Figure 12 shows that a significant amount of pollutant has been degraded at 150 min, indicated by the removal of RhB by more than over 95% for the case of the composite Ag-TiO_2_/microgel. Additionally, it is revealed that the produced composites are superior to Ag-TiO_2_ NPs regarding their photocatalytic activity under visible-light irradiation conditions, and this superiority was statistically significant, according to the Kruskal–Wallis test. This might be because the microgel could act as a bridging mean, keeping the photocatalyst closer to the pollutant compared with the case of its absence. However, pure Ag-TiO_2_ can also degrade the pollutant, but not as efficiently. Both Ag-TiO_2_ and Ag-TiO_2_/microgel show statistically significantly different behavior than the microgel itself, or the light itself (photolysis sample) leaves the pollutant RhB totally unaffected.

First-order kinetics, which were obtained through the Langmuir–Hinshelwood equation, were normalized for the reactions that occurred at the liquid–solid interface, following a standard methodology, which was previously described [18,32,75] (Figure 12b).

For the colored compounds, such as pigments, inks and dyes, their degradation rate commonly increases as the dye concentration increases, but as soon as it attains a critical specific concentration level, it begins to decrease. This reduction in the degradation rate with the increasing concentration of the pollutant could be attributed to the screening of UV-visible-light radiation by the dye molecules, before finally reaching the catalyst surface. Actually, the catalyst concentration can be adjusted for the concentration of the organic compound, so that it can be considerably adsorbed on the photocatalyst surface and thus efficiently degraded [76]. In order to be in accordance with the typical concentrations of the pollutants that are found in real wastewater, the common degradation studies also apply similar concentrations of the organic dyes, in the range of 10–200 mg/L. The same strategy was employed in the present study.

Table 1 gathers the evaluated photoinduced degradation rate constant (k_app_) and the coefficient R^2^ of the linear regression that fit for all the tested materials. As shown through the R^2^ values, the linear kinetic model seems to satisfactorily fit in all the experimental data. The composite material exhibits the best photocatalytic performance. The intrinsic absorbance of RhB dye measured at 554 nm was used to calculate the removal efficiency by using Equation (1):(1)$$\mathrm{Removal}\;\mathrm{efficiency}\;\left(\%\right)=\frac{C_{0}-C}{C_{0}}\cdot 100$$

#### 3.2.2. Effect of Radical Scavengers—Proposed Reaction Mechanism

In order to investigate the main reactive species, including h^+^, ·OH and ·O_2_^−^, which were produced during the photocatalytic degradation of RhB, the trapping series of experiments were implemented in the presence of the radicals’ scavengers, determining the photocatalytic mechanism of the Ag-TiO_2_ NPs. Thus, ethylenediaminetetraacetic acid disodium salt (EDTA-2Na), benzoquinone (BQ) and isopropanol (IPA) were introduced, acting as scavengers, to test the effect of ·OH, ·O_2_^−^ and h^+^ active species in the photocatalytic experiment, respectively [77,78] (Table 2).

The results demonstrated that the photocatalytic degradation efficiency of Ag-TiO_2_ NPs decreased in the presence of scavengers, and this decrease is statistically significant [73,78] (Figure 13). The photocatalytic studies with scavengers were performed under the same test conditions as those previously mentioned in the absence of them (see Section 3.2.1). The concentration of RhB/scavenger solutions were set at 0.001 M [53,78]. In Table 3, the calculated photoinduced degradation rate constant (k_app_) and the R^2^ coefficient of the linear regression are presented, fitting for all the scavengers.

The presence of IPA led to a statistically significant reduction in the photodegradation, indicating hydroxyl radical as the dominant ROS species [53,78]. The photocatalytic degradation efficiency of Ag-TiO_2_ was inhibited more when EDTA-2Na was used, proving that the most reactive species of prepared Ag-TiO_2_ in the photodegradation process were h+ (Figure 14). In IPA trapping experiments suggested that ·OH was another important photocatalytic active agent in Ag-TiO_2_. By contrast, the function of ·O_2_^−^ is the lowest in the presence of all the reactive agents, and it was obtained from the trapping experiments [77,79].

#### 3.2.3. Photocatalytic Mechanism

Photocatalysis is a quite complex process. Generally, the photocatalytic reaction depends on the wavelength, the light (photon) energy and the type of the catalyst. The photocatalytic mechanism includes the following steps, according to the literature [8,13,17]:(a)Upon irradiation with an appropriate light source, providing energy at minimum equal to the value of the photocatalyst E_g_, the electrons that exist on the VB are agitated and finally move to the CB of the semiconductor. As a consequence, positively charged holes are left in the VB of the semiconductor, oxidizing donor molecules and reacting with the available water molecules in order to generate hydroxyl radicals that have strong oxidizing potential, which is capable of degrading various pollutants or damaging or killing biomolecules, leading cells to undergo apoptosis.(b)The electrons of the CB are ready to react with the nearby dissolved oxygen species, forming superoxide ions; thus, these electrons can induce and mediate the redox reactions.(c)The electrons and the produced holes undergo consequent oxidation and reduction reactions with any species that are adsorbed on the semiconductor surface, giving the necessary products as separate for a short time.

According to a research report [75], depending on the reaction conditions, the holes and the ·OH radicals, ·O_2_^−^, H_2_O_2_ and O_2_ play crucial roles during the photocatalytic reaction process. Especially, the increased photocatalytic activity of Ag-TiO_2_ NPs against RhB dye could be easily explained by the SPR effect of Ag and by the visible-light absorption by RhB on the catalyst surface (Figure 15). The SPR of Ag is responsible for the production of electrons and holes (separated charges) in the presence of visible light. Again, RhB*, excited by visible-light irradiation, can transfer electrons to the CB of the semiconductor (TiO_2_) and thus lead to the generation of RhB^+^, which is then degraded through oxidation. Additionally, the electrons (CB electrons of TiO_2_) are transferred to Ag particles. The electron that is generated by the SPR of Ag and the transferred electron from the CB of TiO_2_ (from RhB*) are absorbed by the O_2_, which can further produce ·O_2_^−^. The superoxide radicals (·O_2_^−^) further generate ·OOH, ·OH and ·H_2_O_2_, which can degrade the dye [13,14]. Figure 16 schematically represents the photocatalytic degradation of RhB, by Ag-TiO_2_ NPs, under visible-light irradiation.

### 3.3. Biological Effect

#### 3.3.1. Effect on Cell Proliferation

MDA-MB-231 (cell line with metastatic profile), MCF-7 (low metastatic potential) and HEK293 (normal cells) were treated with increasing concentrations of Ag-TiO_2_ NPs, PNIPAM/PAA salt and PNIPAM/PAA microgel/Ag-TiO_2_. In particular, the concentration of the dispersions that was tested was selected as being in the range of 0–0.75 mg/mL. Additionally, the counting of the cells allowed the creation of growth rates, estimating the proliferation rate. There was no significant effect on the cell proliferation of HEK 293 cells in the presence of any of the produced material (Figure 17a–c), even after visible-light irradiation; thus, normal cells were left unaffected. Various other studies have indicated that primary rat hepatocytes, human lung fibroblasts and other types of normal cells were not affected by TiO_2_ NPs [12]. Moreover, Ag-TiO_2_ NPs, PNIPAM/PAA salt and PNIPAM/PAA microgel/Ag-TiO_2_ before and after irradiation had no effect on MCF-7 cells, whereas *cis*platin significantly decreased this cell population (Figure 18a–c).

These findings are in accordance with some of our previous studies [3,32,43,44,45]. PNIPAM/PAA salt did not also affect the MDA-MB-231 cells (Figure 19a); thus, PNIPAM/PAA salt is not cytotoxic in any of the cell lines in this range of concentrations (Figure 17a, Figure 18a and Figure 19a). It is well known that PNIPAM and various copolymers based on PNIPAM are biocompatible materials, so our findings are commensurate with previous studies on other biological systems [80]. In the presence of Ag/TiO_2_ NPs, MDA-MB-231 cells seem to be functional (Figure 19b). It is clear that the cell proliferation of the MDA-MB-231 cells was decreased when 0.75 mg/mL of photoexcited Ag/TiO_2_ NPs were added (Figure 19b). Hence, a cell-dependent toxicity was observed [32,43,45], perhaps due to the different interactions between the photoactivated materials and cell membranes. Several studies support a similar finding in the anticancer behavior of Ag/TiO_2_ NPs, such as that of Ahamed et al., who found that A549 human pulmonary cancer cells were unaffected by Ag/TiO_2_ NPs, whereas there was significant toxicity in human liver cancer (HepG2) cells [12]. Photoactivated PNIPAM/PAA microgel/Ag-TiO_2_ NPs were more effective and reduced the cell population of MDA-MB-231 cells (Figure 19c). The embedment of Ag-TiO_2_ in the biocompatible microgel increased the efficacy of the nanoparticles to induce cell death or even to inhibit cell proliferation. The cell population was 50% lower in the presence of 0.50 mg/mL of photoexcited PNIPAM/PAA microgel/Ag-TiO_2_ NPs. Perhaps the microgel facilitates the release of Ag-TiO_2_ NPs close to cell membranes and favors the local photocatalytic action of Ag-TiO_2_ NPs [32]. Thus, the polymeric microgel is actually enhancing the photocatalytic efficiency of Ag-TiO_2_ NPs.

The selection of the tested materials was based partly on our previous systematic studies focusing on the titania-based materials (TiO_2_ [43,44], N/TiO_2_ [3,32] and Ag/TiO_2_ NPs [45]), optimizing their performance by chemically modifying TiO_2_ with silver and embedding them inside a thermoresponsive microgel. Thus, among several photocatalysts, TiO_2_ remains a promising biocompatible material, and for this reason, Ag/TiO_2_ was developed.

Because the ultimate goal of this research, and for our future plans, is to develop a drug-delivery system that can efficiently kill cancer cells, a polymeric material was critical to be added to the composite particles to provide the required properties and in particular to transfer the material to the target area. Thus, various materials could be used, such as liposomes [81], poly (N-vinyl caprolactam) (PNVCL) [82], chitosan [83] or PEO-b-PCL–DPPC chimeric nanocarriers [84,85]. PNIPAM/PAA microgel combines the required properties of the aforementioned materials, allowing the development of a material with a controlled size [41,42] that can adequately act at 37 °C, a temperature close to that of a human body, and therefore was considered among the most promising biopolymers to be used in our study in order to fulfill our future plans of designing a photoinduced targeted drug-delivery system with anticancer performance.

#### 3.3.2. Effect on Cytotoxicity

MTT colorimetric assay was applied to investigate the viability of HEK293, MCF-7 and MDA-MB-231 cells, as the concentration of Ag-TiO_2_ NPs, PNIPAM/PAA salt and PNIPAM/PAA microgel/Ag-TiO_2_ NPs increased before and after of their photoactivation with visible light. By using well-known MTT protocols, the cell viability was estimated as a percentage ratio of the measured optical density of the treated cells to untreated ones. All the types of cells remained unaffected in the presence of PNIPAM/PAA salt or photoactivated PNIPAM/PAA salt, meaning that this material was not cytotoxic, at least in this range of concentrations (Figure 20a,b). Indeed, PNIPAM/PAA and other stimuli-responsive polymers are considered as suitable materials for biomedical applications, as drug carriers [86]. Ag-TiO_2_ NPs did not induce any cytotoxic effect on these cell lines (Figure 20c). The cell population was also unaffected; thus, this finding was in agreement with that of growth rates (Figure 17b, Figure 18b and Figure 19b). Photoactivated Ag-TiO_2_ NPs had no biological effect on HEK293 and MCF-7 (Figure 20d), whereas a gradual decrease in cell viability of MDA-MB-231 cells was revealed (Figure 20d). In particular, 0.5 mg/mL of photoactivated Ag/TiO_2_ NPs reduced MDA-MB-231 cell viability by 20% and 0.75 mg/mL by 30%. As is clear in Figure 16e, cell viability did not change in the presence of PNIPAM/PAA microgel/Ag-TiO_2_ NPs for all the cell lines. Photoactivated PNIPAM/PAA microgel/Ag-TiO_2_ NPs had no effect on the cell viability of HEK293 and MCF-7 cells (Figure 20f). Interestingly, under photoenhancement with visible light, the cell viability of MDA-MB-231 cells gradually decreases (Figure 20f). Even 0.25 mg/mL can reduce cell viability by 20%, and 0.75 mg/mL of the composite material finally induced a 50% decrease in cell viability (IC50). Hence, embedment in this thermoresponsive microgel increased the photocatalytic efficiency of Ag-TiO_2_ NPs. Thus, a cell-type selectivity was observed, where MDA-MB-231 was more vulnerable to these materials.

In order to further develop a drug-delivery system that can be efficient for in vivo treatments, it is crucial to design a 3D tissue model [87]. Our cell-line-based experiments are very promising, but in order to predict the behavior of the produced materials in vivo [88], adding an intermediate step using a 3D model (probably an alginic/gelatin-based scaffold-like system or organoids) is among our future plans.

## 4. Conclusions

Nanostructured Ag/TiO_2_ was produced via the sol-gel synthesis method and embedded within PNIPAM/PAA microgel, creating a thermoresponsive composite material. Full characterization, employing XRD, micro-Raman, FT-IR, UV-Vis, DLS and TEM confirmed the physicochemical properties and the morphology of the produced material, verifying the chemical modification of TiO_2_ with silver and the successful development of the composite material. The photocatalytic efficacy of the produced Ag-TiO_2_ was broadened, including also the visible-light range of the electromagnetic spectrum, thanks to doping with silver in that the produced Ag-TiO_2_ nanopowder revealed that there was a decrease in the energy band gap E_g_ (E_g_ = 2.32 eV) when it was compared with pure anatase (E_g_ = 3.1 eV). This decrease indicated that Ag-TiO_2_ had better photocatalytic activity under irradiation with visible light, because the chemically modified catalyst finally needed lower energy for its photoactivation. Anatase was found to be the dominant crystal phase of the Ag-TiO_2_ powders, as XRD and Raman analysis results displayed, while the composition of the produced materials was confirmed via FT-IR analysis. The average crystallite size of the produced nanoparticles was estimated to be in the nanoscale, particularly 9.52 nm, as it was obtained through XRD and their zeta potential (ZP = (−53.5 ± 6) mV), indicating their stability. The polymeric network was synthesized following the steps of a precipitation polymerization process, presenting a VPTT at 37.6 °C, which allows the controlled release of a pharmaceutic factor, such as TiO_2_, inside a biological system. Afterward, the Ag-TiO_2_ NPs were successfully embedded in the network (as indicated in TEM images). Hence, embedding the nanoparticles resulted in the formation of the composite material (PNIPAM/PAA microgel/Ag-TiO_2_ NPs), which could act as a thermoresponsive and innovative drug-delivery system.

Photocatalytic tests using RhB pollutant reassured the visible-light activation of the powders, with the composite material succeeding to almost totally degrade the RhB pollutant (over 95%), after 150 min of irradiation with visible light. Composite nanoparticles demonstrated better photocatalytic activity compared with Ag-TiO_2_ NPs (85.3%), as we expected, because of the structure of microgels. Microgels act like sponges of high porosity that trap the photocatalytic nanoparticles close to the pollutant. In this way, the photocatalytic reaction can take place faster, and the degradation of the pollutant is achievable. The implementation of the same photocatalytic experiments was implemented in the presence of radicals’ scavengers (ethylenediaminetetraacetic acid disodium salt (EDTA-2Na), benzoquinone (BQ) and isopropanol (IPA)), indicating the hydroxyl radical as the dominant reactive oxygen species, shedding light on the mechanism through which this photocatalyst acts.

Furthermore, for the analysis of their anticancer behavior, two breast cancer epithelial cell lines (MCF-7 and MDA-MB-231) and normal human embryonic kidney cells (HEK 293) were cultured and treated with the produced materials (the concentration was in the range of 0–0.75 mg/mL), under irradiation with visible light, and cell proliferation and cytotoxicity assays were employed. All the tested materials did not affect the cell proliferation rate or the cell viability before their photoactivation. This was a very promising result in that it is important to control the anticancer potential of the developed materials. Under visible-light irradiation, PNIPAM/PAA was still harmless for all the cell lines, while Ag-TiO_2_ NPs gradually decreased the cell proliferation and the MDA-MB-231 cell viability, leaving the MCF-7 and HEK293 cells unaffected. An amount of 0.50 mg/mL of photoactivated PNIPAM/PAA microgel/Ag-TiO_2_ NPs decreased the cell population of MDA-MB-231 cells by 50%.

Moreover, photoactivated PNIPAM/PAA microgel/Ag-TiO_2_ NPs had a more significant killing effect on MDA-MB-231 cells, with the IC50 being 0.75 mg/mL for this cell line, while at the same concentration, the cell viability of the MDA-MB-231 cells was decreased by 25–30% in the presence of Ag-TiO_2_ NPs, meaning that the composite material had enhanced anticancer efficiency.

Further clarification on the interactions between the different cell types would be in our future plans because a cell-dependent toxicity was observed. These promising findings will encourage us to continue to optimize our method in order to develop an alternative photodynamic cancer therapy, one that is based on a drug-delivery system that allows the controlled release of the photocatalytic anticancer material at the target area, maximizing the therapeutic effect on the cancer cells and avoiding the harmful side effects on healthy tissue, while reducing costs to the healthcare system supporting those treatments.

## Figures and Tables

**Figure 1 pharmaceutics-15-00135-f001:**
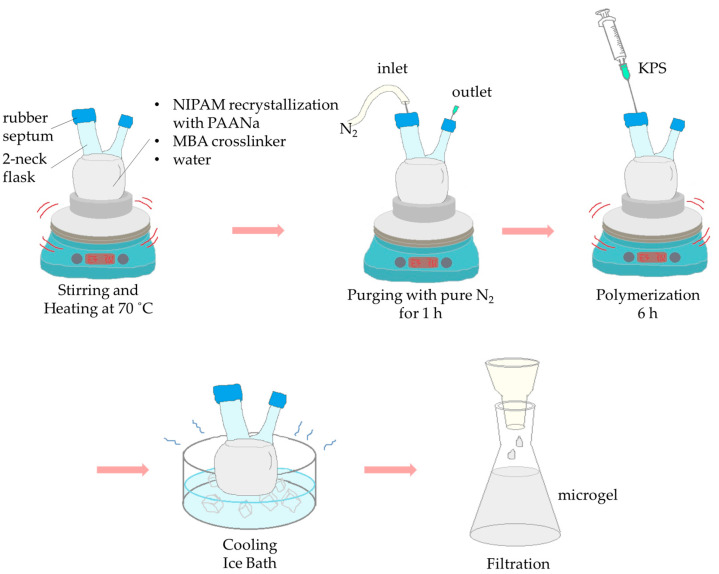
Schematic representation of the synthesis of PNIPAM/PAA microgel by surfactant-free precipitation polymerization.

**Figure 2 pharmaceutics-15-00135-f002:**
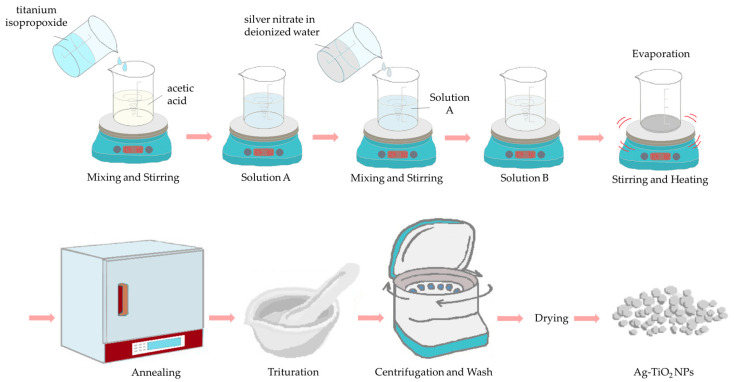
Schematic representation of the synthesis of Ag-TiO_2_ nanoparticles.

**Figure 3 pharmaceutics-15-00135-f003:**
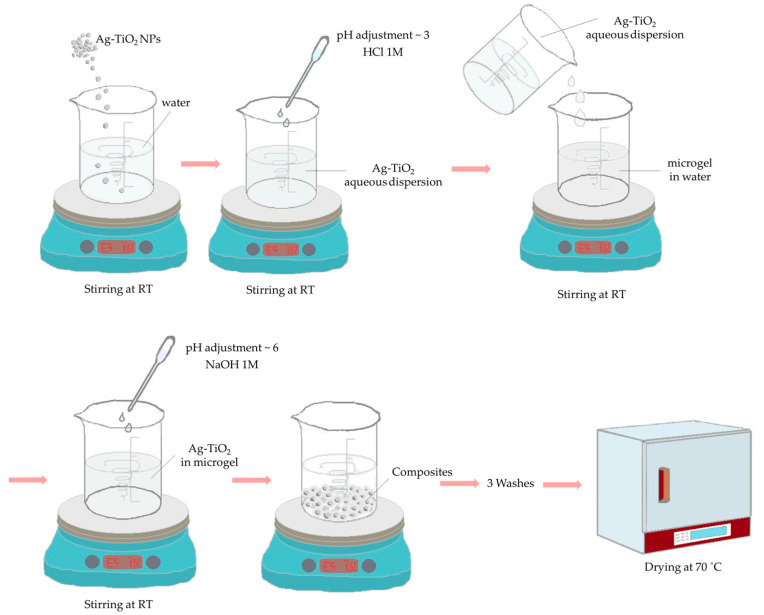
Schematic representation of the synthesis of composite nanoparticles.

**Figure 4 pharmaceutics-15-00135-f004:**
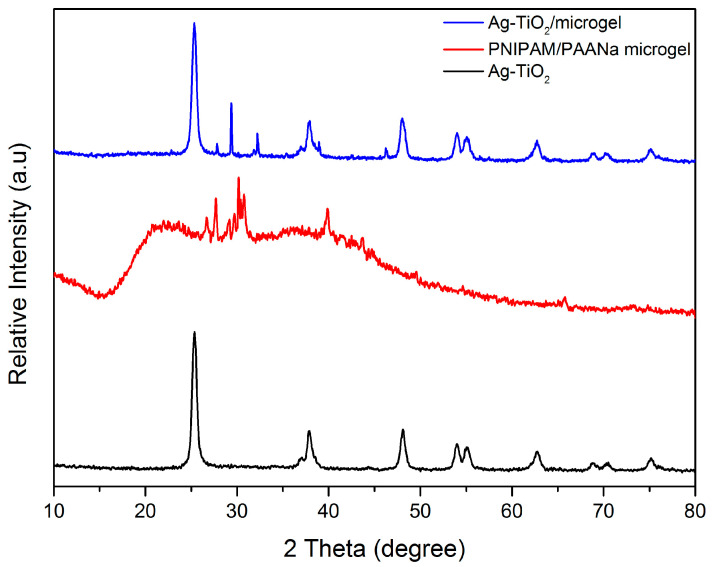
XRD patterns of Ag-TiO_2_ nanoparticles (in black), PNIPAM/PAA salt (in red) and PNIPAM/PAA microgel/Ag-TiO_2_ (in blue).

**Figure 5 pharmaceutics-15-00135-f005:**
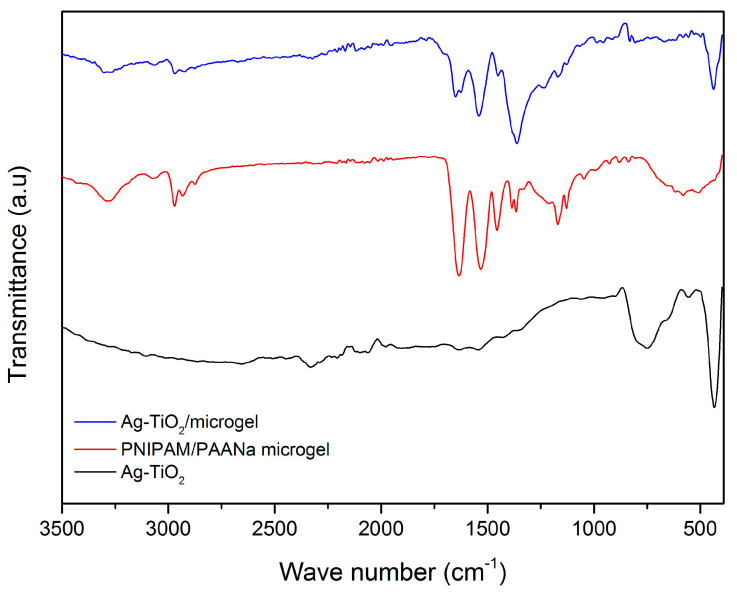
FT-IR spectra of Ag-TiO_2_ nanoparticles (in black), PNIPAM/PAA salt (in red) and PNIPAM/PAA microgel/Ag-TiO_2_ composite (in blue).

**Figure 6 pharmaceutics-15-00135-f006:**
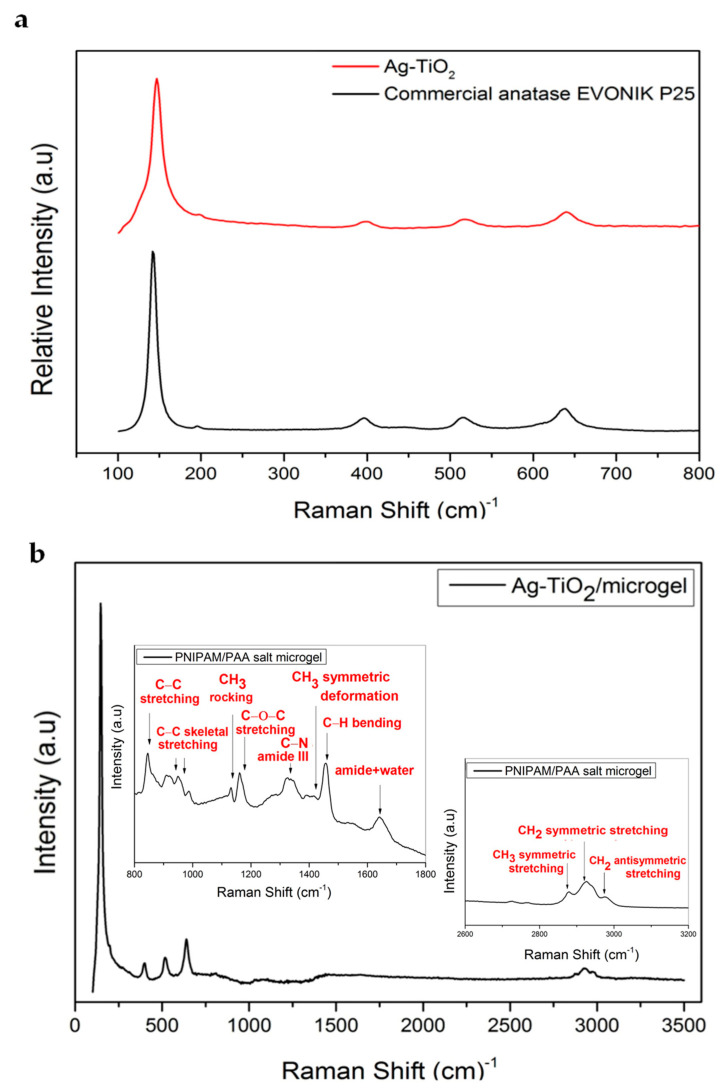
Raman spectra of (**a**) Ag-TiO_2_ nanoparticles compared with commercially available Evonik P25 TiO_2_ anatase and (**b**) PNIPAM/PAA microgel/Ag-TiO_2_. The insets depict a part of the Raman spectrum of PNIPAM/PAA salt microgel.

**Figure 7 pharmaceutics-15-00135-f007:**
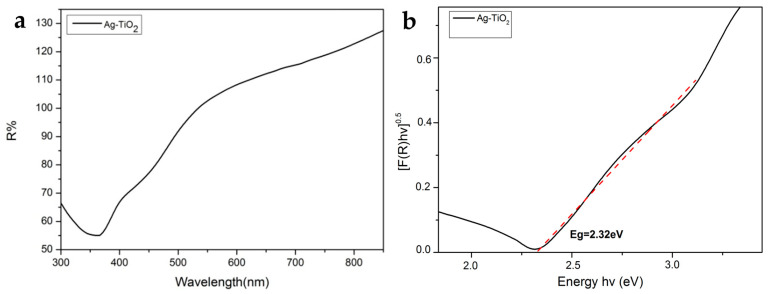
(**a**) F(R) reflectance as a function of wavelength for Ag-TiO_2_ nanoparticles. (**b**) Optical energy band gap (E_g_) of the Ag-TiO_2_ nanoparticles.

**Figure 8 pharmaceutics-15-00135-f008:**
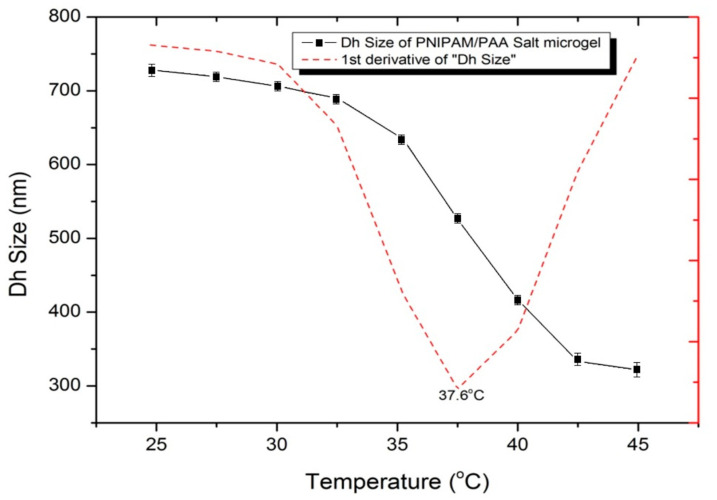
Size (D_h_—hydrodynamic diameter) of the microgel PNIPAM/PAA salt, as a function of temperature through DLS. The suspension pH was equal to 7.4. The red dash line underlines the VPTT, which corresponds to 37.6 °C. The data represent means ± standard deviation from three experiments.

**Figure 9 pharmaceutics-15-00135-f009:**
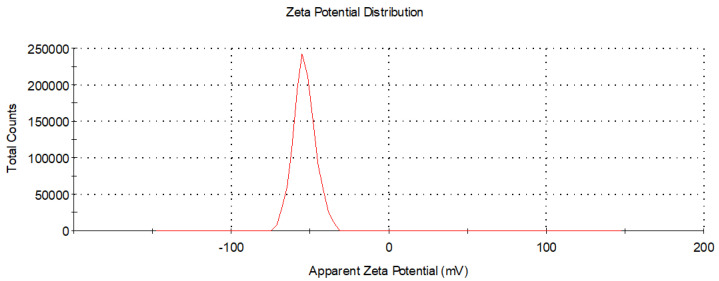
Zeta potential distribution of Ag-TiO_2_ nanoparticles via DLS. The suspension pH was equal to 7.4.

**Figure 10 pharmaceutics-15-00135-f010:**
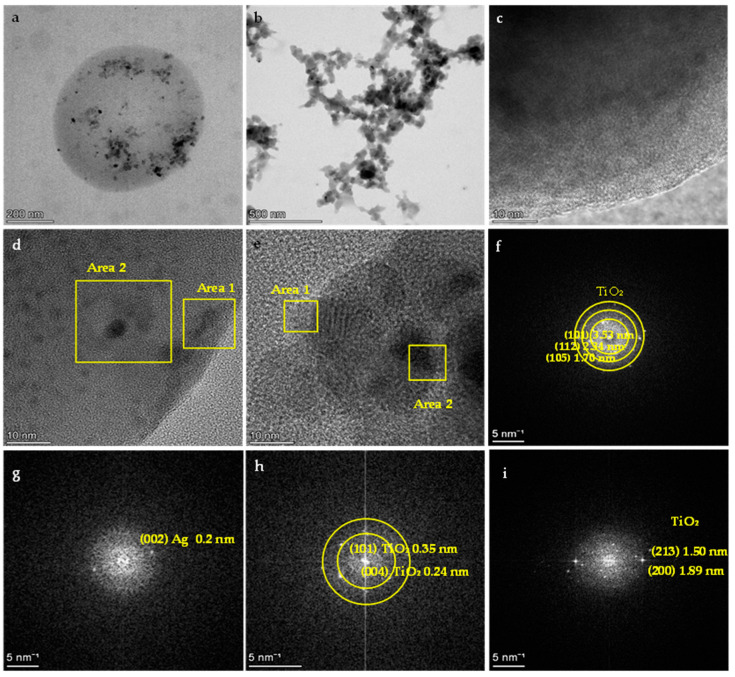
Transmission electron microscopy images of (**a**) the spherical particle of microgel with embedded Ag-TiO_2_ nanoparticles, (**b**) the PNIPAM/PAA microgel Ag-TiO_2_ NPs, (**c**) the outline of a microgel particle, (**d**) a microgel particle with embedded Ag-TiO_2_ nanoparticles and lattice measurement spacing, (**e**) embedded Ag-TiO_2_ nanoparticles with lattice measurement spacing, (**f**) TiO_2_ nanoparticles, (**g**) Ag nanoparticles and (**h**,**i**) TiO_2_-decorated Ag nanoparticles.

**Figure 11 pharmaceutics-15-00135-f011:**
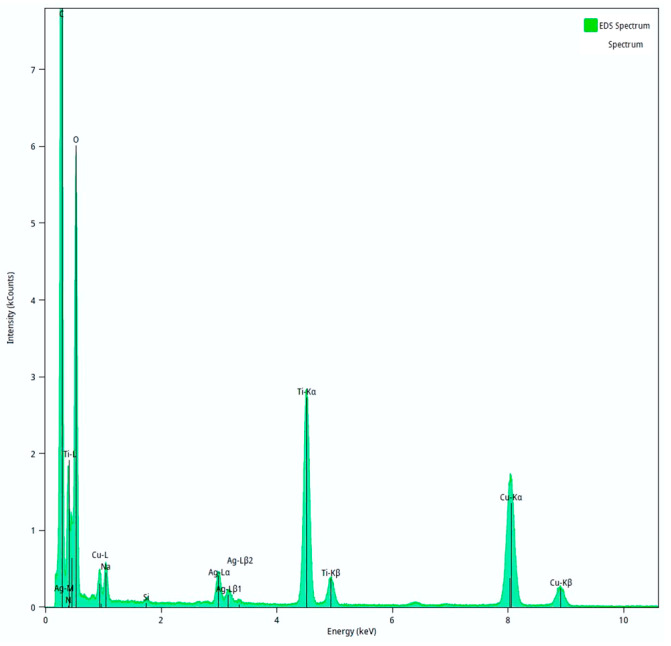
Energy-dispersive spectroscopy (EDS) analysis of Ag-TiO_2_/microgel composite.

**Figure 12 pharmaceutics-15-00135-f012:**
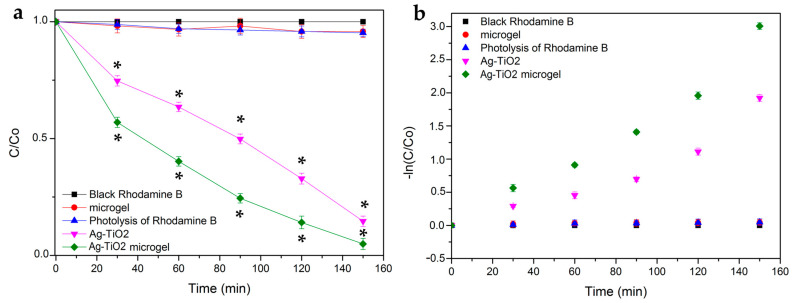
(**a**) Degradation curves of rhodamine B for Ag-TiO_2_, PNIPAM-co-PAA microgel and PNIPAM -co-PAA Ag-TiO_2_ composite as a function of time, under visible-light irradiation. The phenomenon of rhodamine B photolysis and its degradation in the dark are also studied. The data represent means ± standard deviation from three experiments. In these series of experiments, * *p* < 0.05 was considered statistically significant. (**b**) Photocatalytic kinetics of the degradation of rhodamine B for Ag-TiO_2_, PNIPAM-co-PAA microgel and PNIPAM-co-PAA Ag-TiO_2_, according to a linear pseudo-first-order model.

**Figure 13 pharmaceutics-15-00135-f013:**
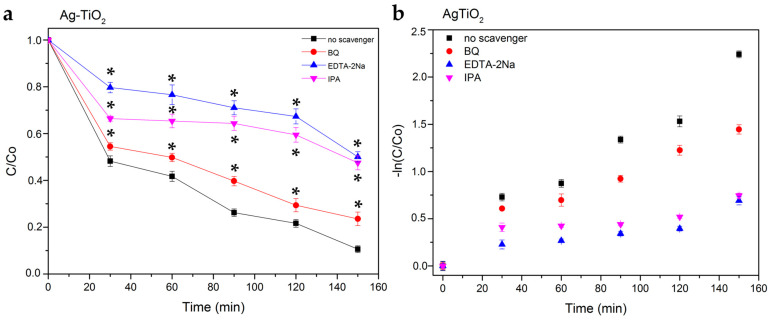
(**a**) Photocatalytic degradation curves of RhB in the presence of Ag-TiO_2_ and the radicals’ scavengers (ethylenediaminetetraacetic acid disodium salt (EDTA-2Na), benzoquinone (BQ) and isopropanol (IPA)). The data represent means ± standard deviation from three experiments. In these series of experiments, * *p* < 0.05 was considered statistically significant. (**b**) Photocatalytic kinetics of the same samples, according to a linear pseudo-first-order model.

**Figure 14 pharmaceutics-15-00135-f014:**
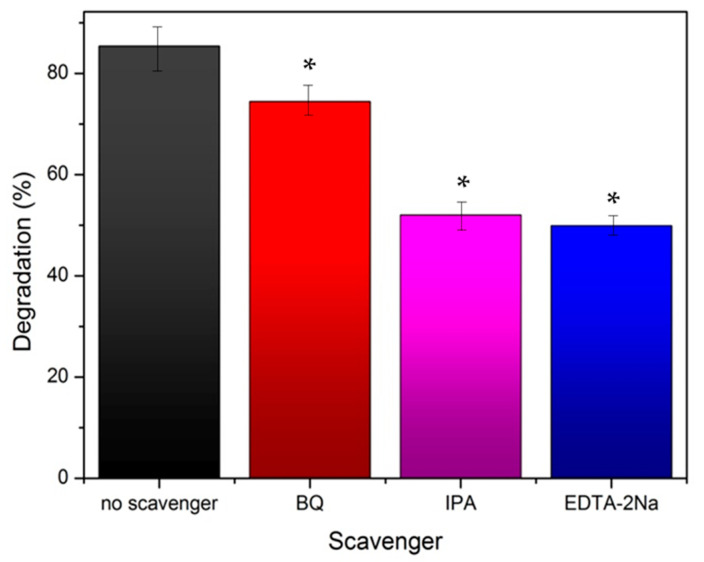
Percentage of degradation of rhodamine B by Ag-TiO_2_ nanoparticles (in black) and also in the presence of scavengers: ethylenediaminetetraacetic acid disodium salt (EDTA-2Na) (in blue), benzoquinone (BQ) (in red) and isopropanol (IPA) (in purple). The data represent means ± standard deviation from three experiments. In these series of experiments, * *p* < 0.05 was considered statistically significant.

**Figure 15 pharmaceutics-15-00135-f015:**
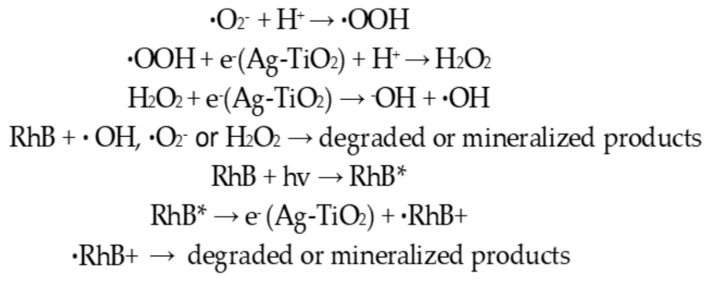
Photocatalytic degradation reactions of RhB.

**Figure 16 pharmaceutics-15-00135-f016:**
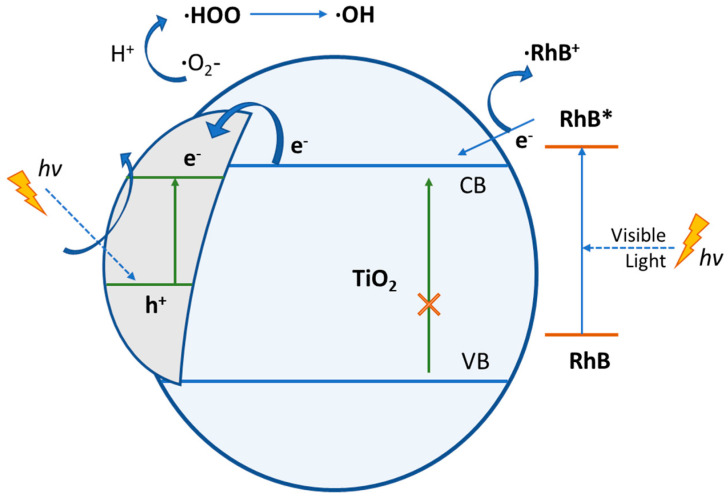
Schematic representation of the photocatalytic degradation of RhB by Ag–TiO_2_ NPs, under visible-light irradiation.

**Figure 17 pharmaceutics-15-00135-f017:**
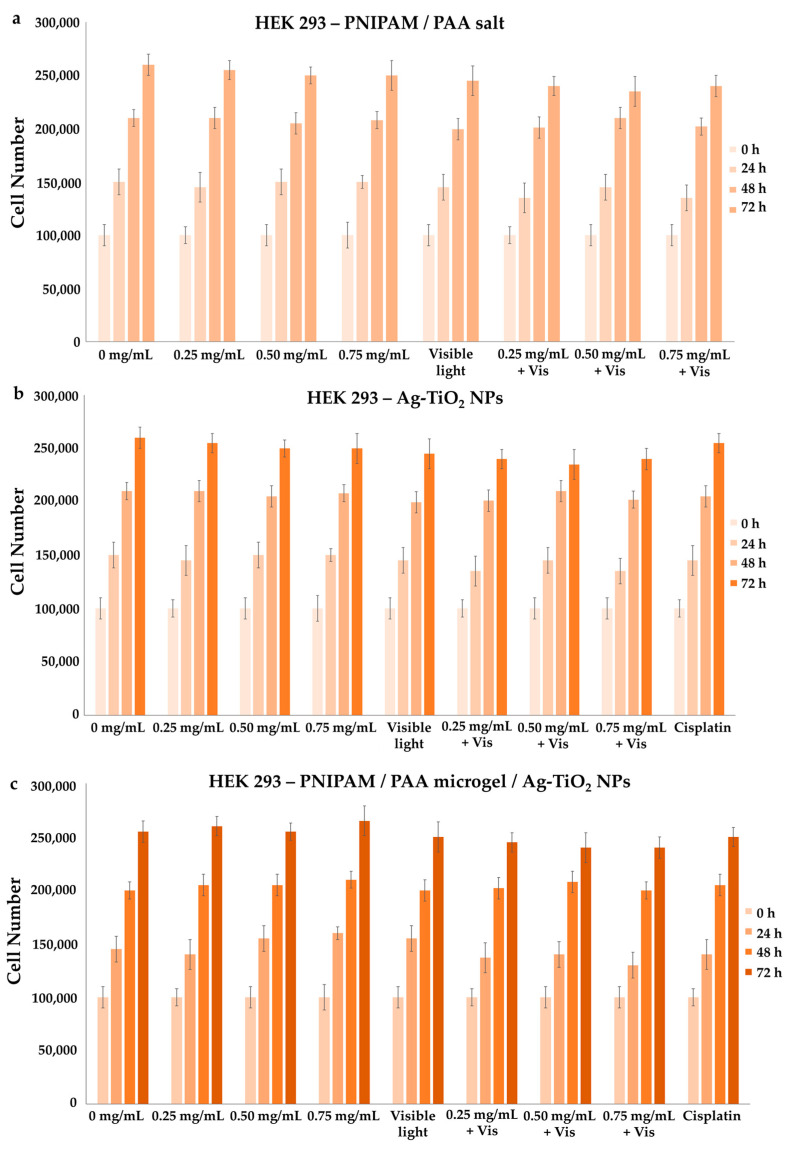
Growth rates of HEK293 cell line, with the addition of (**a**) PNIPAM/PAA salt, (**b**) Ag-TiO_2_ NPs and (**c**) PNIPAM/PAA microgel/Ag-TiO_2_. There was no effect on cell proliferation, as the concentration of each of the tested materials increased. Additionally, irradiation with visible light did not affect their proliferation. In these series of experiments, *p* < 0.05 was considered statistically significant.

**Figure 18 pharmaceutics-15-00135-f018:**
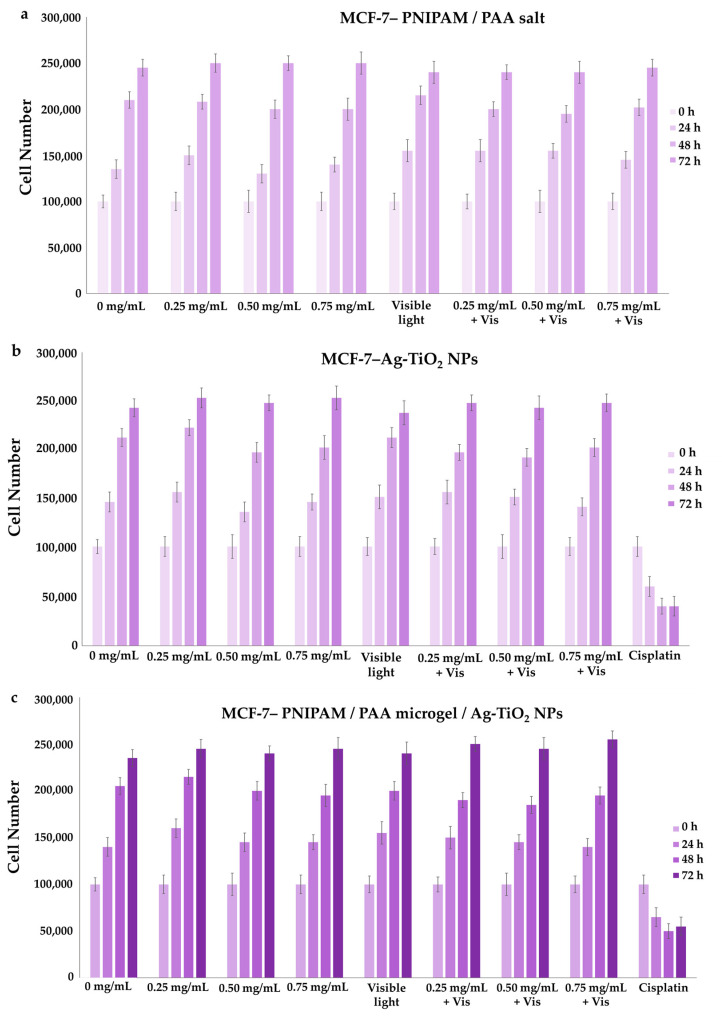
Growth rates of MCF-7 cell line, with the addition of (**a**) PNIPAM/PAA salt, (**b**) Ag-TiO_2_ NPs and (**c**) PNIPAM/PAA microgel/Ag-TiO_2_. There was no detectable effect on cell proliferation in the presence of each of the produced materials in increasing concentrations, even after their photoactivation. Samples treated with *cis*platin were used as positive controls in the experiment. In these series of experiments, *p* < 0.05 was considered statistically significant.

**Figure 19 pharmaceutics-15-00135-f019:**
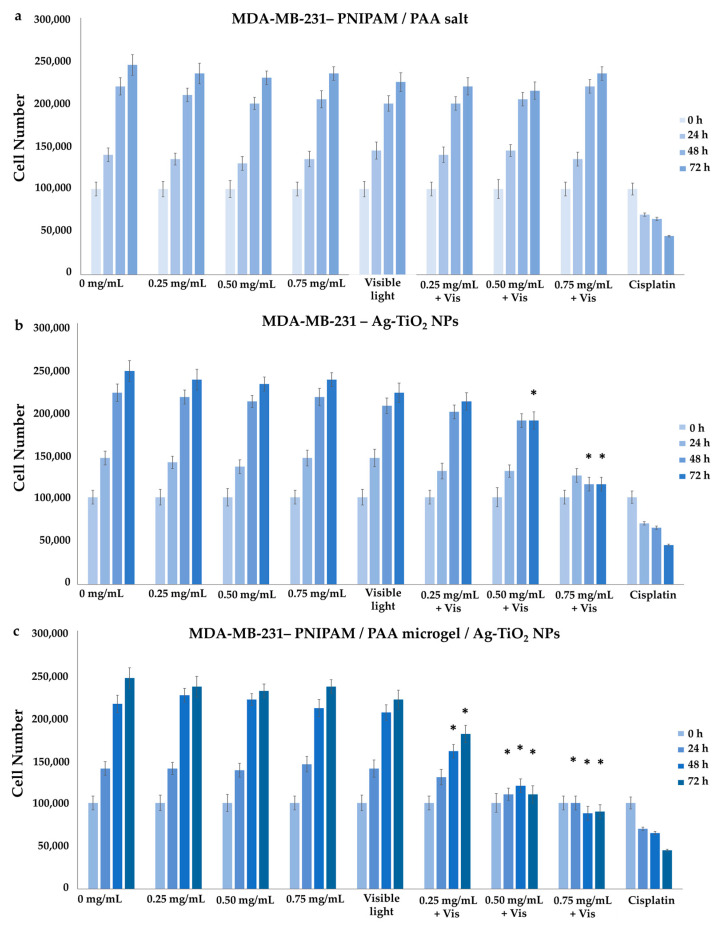
Growth rates of MDA-MB-231 cell line, with the addition of (**a**) PNIPAM/PAA salt, (**b**) Ag-TiO_2_ NPs and (**c**) PNIPAM/PAA microgel/Ag-TiO_2_. There was no effect on cell proliferation in the presence of increasing concentrations of PNIPAM/PAA salt or Ag-TiO_2_ before irradiation. Under irradiation with visible light, there was a decrease in cell population when 0.75 mg/mL of Ag/TiO_2_ NPs were added. The photoexcitement of PNIPAM/PAA microgel/Ag-TiO_2_ NPs resulted in a significant decrease in this cell proliferation. Samples treated with *cis*platin were used as positive controls in the experiment. In these series of experiments, * *p* < 0.05 was considered statistically significant.

**Figure 20 pharmaceutics-15-00135-f020:**
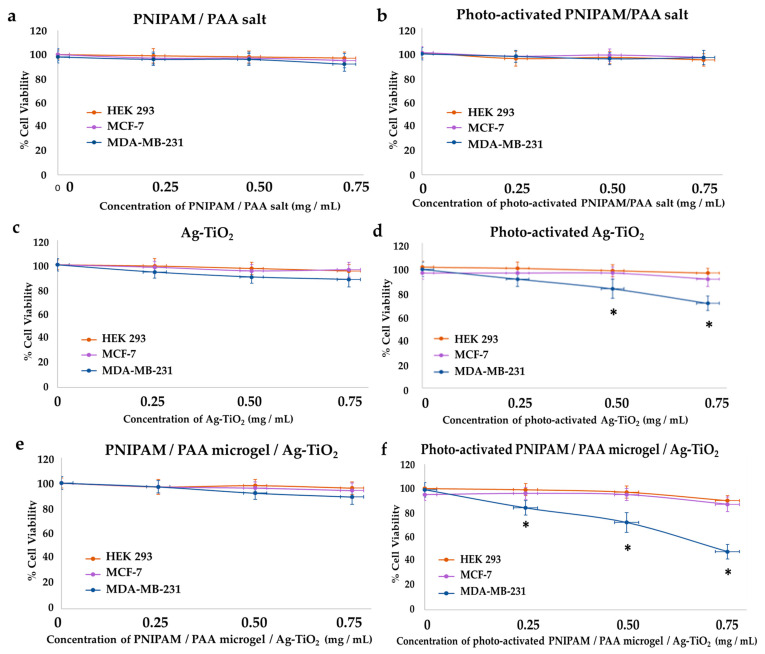
Effect of the produced materials on cell viability. MTT colorimetric assay was employed to estimate the percentage of cell viability of HEK293, MCF-7 and MDA-MB-231 cells in the presence of PNIPAM/PAA salt (**a**,**b**), Ag-TiO_2_ NPs (**c**,**d**) and PNIPAM/PAA microgel/Ag-TiO_2_ NPs (**e**,**f**) in increasing concentrations and also before and after photoactivation with visible light. Photoactivated Ag-TiO_2_ NPs gradually decreased the cell viability of MDA-MB-231 cells, and this phenomenon was more intense in the presence of PNIPAM/PAA microgel/Ag-TiO_2_ NPs. * *p* < 0.05 vs. negative control, through the Kruskal–Wallis nonparametric test. The obtained data represent means ± standard deviation from four experiments.

**Table 1 pharmaceutics-15-00135-t001:** R^2^—linear correlation coefficients; k_app_—photocatalytic degradation rate constant, for the produced materials and pure rhodamine B solution.

Material	R^2^	Kapp (min^−1^)	Degradation %
Microgel	0.86271	2.96 × 10^−4^	4.1%
Rhodamine B	0.95107	3.37 × 10^−4^	4.76%
Ag-TiO_2_	0.91016	1.13 × 10^−2^	85.3%
Ag-TiO_2_/microgel	0.96734	1.84 × 10^−2^	95%

**Table 2 pharmaceutics-15-00135-t002:** Reactive oxygen species and h+ can be trapped by various scavengers.

Scavenger	ROS
EDTA-2Na	hole (h^+^)
BQ	superoxide radicals(·O_2_^−^)
IPA	hydroxyl radical (·OH)

**Table 3 pharmaceutics-15-00135-t003:** Linear correlation coefficients (R^2^) and photocatalytic degradation rate constant (k_app_) for all the scavengers.

Ag-TiO_2_	R^2^	Kapp (min^−1^)
EDTA-2Na	0.87798	3.85 × 10^−3^
BQ	0.94117	8.86 × 10^−3^
IPA	0.76272	3.88 × 10^−3^

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
