# Peer review of "Composite Nanoarchitectonics of Photoactivated Titania-Based Materials with Anticancer Properties"

_pharmaceutics, 2022, doi:10.3390/pharmaceutics15010135_

Round 1

Reviewer 1 Report

This research is focused on the photocatalytic activity of titania-based composite material for dye pollutant degradation and cancer treatment via photoactivation. Authors have synthesized Ag-TiO2 NPs using the sol-gel method followed by characterization using XRD, FT-IR, micro-Raman Spectroscopy, Ultra-Violet-Visible spectroscopy, DLS, and TEM. There are multiple technical flaws found regarding the Experimental design. The article has many grammatical and sentence errors, and the language organization needs to be improved. For these reasons, I conclude that the paper suitable for publication with a major revision

1.      Authors have mentioned Thermo-responsive material in the title but photoactivated material-based studies have been studied.

2.      Title does not reflect the various data provided in the manuscript.

3.      Please make sure that all keywords have been used in the abstract and the title.

4. The preparation and synthesis section does not contain enough information to reproduce. Protocols need to be rewritten in detail.

5.      Photocatalytic experiments need to be conducted in the triplets. Mean data need to provide. The concentration of RhB solution needs to provide.

6.      Reusability study for the photocatalyst to understand the economic feasibility of the applications.

7.      Typographical errors can be avoided. The language and grammar used throughout the manuscript need to be improved

8.      Conclusion needs to improve by providing numerical and key results like IC50 value.

9.     The statistical analyses of all monitored data have to be performed and discussed; add ±SD and statistical significance where relevant.

Author Response

1
Response to Reviewer 1 Comments
The authors would like to thank the reviewer for the valuable comments and for giving them the opportunity to revise their manuscript.
They, also, would like to underline that the motivation of our research is to synthesize titania-based composite materials with anticancer potential under visible-light irradiation. Particularly, titanium dioxide (TiO2) nanoparticles (NPs) chemically modified with silver, were embedded in a stimuli-responsive microgel (a cross-linked interpenetrating network (IP) network, that was synthesized by poly (N-Isopropylacrylamide) and linear chains of polyacrylic acid sodium salt), forming composite particles. The ultimate goal of this research and our future perspective is to develop a drug delivery system, based on optical fibers which could photoactivate efficiently NPs, targeting on cancer cells. The produced Ag-TiO2 NPs, the microgel and the composite materials were fully characterized. Our results indicated that Ag-TiO2 NPs were successfully embedded into the thermo-responsive PNIPAM/PAA microgel. Either Ag-TiO2 NPs or the composite materials exhibited a high photocatalytic degradation efficiency on the pollutant Rhodamine B and significant anticancer potential under visible light irradiation.
Since the authors understood the comment of the reviewer about the lack of the title to precisely describe the content of the study, thus the title was redefined.
All the modifications have been highlighted in yellow in the revised version of the manuscript. The reference and the figure numbering have also been changed.
Point 1: Authors have mentioned Thermo-responsive material in the title but photoactivated material-based studies have been studied.
Response 1: The authors considering this comment, reconstructed the title in order to precisely describe the content of their study. Particularly the new proposed title is: “Composite nanoarchitectonics of photo-activated titania-based materials with anticancer properties”
Point 2: Title does not reflect the various data provided in the manuscript.
Response 2: The authors appreciated this comment and changed the title to be sure that accurately describe the content of their study. The new title is: “Composite nanoarchitectonics of photo-activated titania-based materials with anticancer properties”
Point 3: Please make sure that all keywords have been used in the abstract and the title.
Response 3: The authors would like to thank the reviewer for this suggestion. They added two new keywords in order to be in accordance with the new title and the abstract.
Point 4: The preparation and synthesis section does not contain enough information to reproduce. Protocols need to be rewritten in detail.
Response 4: The authors would gratefully thank the reviewer for this remark. They enriched Section «2.1.3. Composite nanoparticles» with additional information. Also, in order to clarify the steps of the synthesis procedure, the authors prepared three additional Figures (Figure 1-3) that schematically represent the whole process.
Point 5: Photocatalytic experiments need to be conducted in the triplets. Mean data need to provide. The concentration of RhB solution needs to provide.
2
Response 5: The authors would like to thank the reviewer for the constructive suggestion. The mean values are presented in the graphs, as well as the error bars obtained from the triplets (Figure 12 and 13). Also, the concentration of RhB solution is 0.01 M and this information has been added in the revised manuscript.
Point 6: Reusability study for the photocatalyst to understand the economic feasibility of the applications.
Response 6: The authors would like to thank the reviewer for this comment. They understand the importance of the reusability of a photocatalyst and the economic impact of applications like this. However, they avoided to add this part in the main manuscript, since the ultimate goal of this study was the development of a drug delivery system that might controllably release inside the tumor, killing cancer cells, thus there is no chance to re-use the same photocatalytic material post administration. Therefore, under this research framework the authors consider that this re-usability study is out of the general scope of this research given that the re-usability issue is not applicable in this therapeutic approach, while it could be in other cases like an environmental applications but not in such case of biomedical aspect.
Actually, the authors have studied the reusability of the produced materials, by collecting the catalyst after the first photocatalytic cycle, centrifugating and filtrating, washing with distilled water and drying at 60 °C in an oven. The Ag-doped TiO2 and Ag-TiO2/microgels have been tested for their photocatalytic performance after 3 cycles, as a part of another scientific project.
The weight loss was determined ~3% of Ag-TiO2 and ~1% of Ag-TiO2/microgel, demonstrating that the microgel protected the aggregation sensitive catalyst due to the network structure, providing a large surface to adsorb pollutants. The authors trust the reviewer, the editor and the journal and they share their results (see Figures 1 and 2, Tables 1 and 2) but not include in the manuscript (the concentration of RhB was 0.01 M).
Figure 1. a) Photocatalytic degradation of RhB after 3 cycles, in the presence of Ag-TiO2 NPs B) Kinetics study.
Table 1. R2: Linear correlation coefficients and kapp: photocatalytic degradation rate constant, for the produced materials and pure Rhodamine B solution.
Material
R2
Kapp (min-1)
Degradation %
Ag-TiO2 v1
0.9101
1.13 · 10-2
85.3
Ag-TiO2 v2
0.8965
1.08 · 10-2
83.6
Ag-TiO2 v3
0.8905
1.0 · 10-2
83
3
Figure 2. a) Photocatalytic degradation of RhB after 3 cycles, in the presence of Ag-TiO2/microgel. B) Kinetics study.
Table 2. R2: Linear correlation coefficients and kapp: photocatalytic degradation rate constant, for the produced materials and pure Rhodamine B solution.
The results indicated negligible effect on kapp values for both produced materials, a enhanced ability to be reused for 3 consecutive cycles, confirming the high recycling performance of these type of catalysts.
The authors believe that even without reusability, the possibility to use such materials could reduce the cost for the health system and for this reason they added a comment in the introduction and the conclusion.
Point 7: Typographical errors can be avoided. The language and grammar used throughout the manuscript need to be improved.
Response 7: The authors considering this comment, tried to correct the detected grammatical errors in the manuscript.
Point 8: Conclusion needs to improve by providing numerical and key results like IC50 value.
Response 8: The authors are thankful for this suggestion. They considered its importance and thus, provided various numerical and key results like IC50 value.
Point 9: The statistical analyses of all monitored data have to be performed and discussed; add ±SD and statistical significance where relevant.
Response 9: The authors would like to thank the reviewer for this valuable and constructive comment. They had already performed statistical analysis for the cytotoxic data, and they also implemented it for the photocatalytic experiments in the revised manuscript.
Statistical analysis was performed, based on the nonparametric Kruskal–Wallis test and p < 0.05 was considered statistically significant. This information is now provided in several points of the manuscript in yellow (e.g., 2.3. Photocatalytic Test, 2.4.3. Cytotoxicity Test, 3.2.1. Photocatalytic Efficiency and Kinetics, 3.2.2. Effect of radical scavengers - Proposed reaction mechanism), as well as in Figures 12, 13 and 14.
Material
R2
Kapp (min-1)
Degradation %
Ag-TiO2/ microgel v1
0.9587
1.87 · 10-2
95
Ag-TiO2/ microgel v2
0.9256
1.78 · 10-2
94.76
Ag-TiO2/ microgel v3
0.9175
1.77 · 10-2
94.5

Reviewer 2 Report

This manuscript actually reports lots of data for various characterizations and properties. Publication of these data in public journal media may have some good contributions to the related research areas. From this positive viewpoint, I may suggest publication of this work in Pharmaceutics. However, the data quality and presentation styles are not in high level. Therefore, entire impression may be poor and routine. Upon revisions on several points, this manuscript becomes better. Please see below.

1) In order to much increase understandability of this work, The used materials and fabrication methods have to be clearly presented. I suggest addition of the first figure to explain the used materials and their fabrication methods.

2) Superiority and/or specific features of the used materials in bio-related properties have to be discussed upon comparisons over the related materials reported in the past literatures. Such discussions would originality and novelty of the work much clearer.

3) Addition of conceptual term to the title would increase innovative impression to this manuscript. I may suggest use of an emerging conceptual term, nanoarchitectonics, in the title (as post-nanotechnology concept, see https://pubs.rsc.org/en/content/articlelanding/2021/nh/d0nh00680g). For example, the title like ... Composite nanoarchitectonics of thermo-responsive titania with photo-induced anticancer properties ... may sound more innovative.

4) Some figures are not in good resolution. Please improve figure quality.

5) In Figure 2, vertical axis would be Transmittance (a.u.) (not Relative Intensity).

6) Please possibly add error bars to plots in several graphs such as Figures 5, 8, and 9.

7) Figure 7 has bad color choice. Green on black background does not have good contrast. Please use lighter color on black backgrounds. Scale bars on these figure cannot be seen well. Please much improve these features.

8) Figure 7f cannot be seen well. This graph can be represented as a separate figure.

Author Response

1
Response to Reviewer 2 Comments
The authors would like to thank the reviewer for the valuable comments and for giving them the opportunity to revise their manuscript.
They, also, would like to underline that the motivation of our research is to synthesize titania-based composite materials with anticancer potential under visible-light irradiation. Particularly, titanium dioxide (TiO2) nanoparticles (NPs) chemically modified with silver, were embedded in a stimuli-responsive microgel (a cross-linked interpenetrating network (IP) network, that was synthesized by poly (N-Isopropylacrylamide) and linear chains of polyacrylic acid sodium salt), forming composite particles. The ultimate goal of this research and our future perspective is to develop a drug delivery system, based on optical fibers which could photoactivate efficiently NPs, targeting on cancer cells. The produced Ag-TiO2 NPs, the microgel and the composite materials were fully characterized. Our results indicated that Ag-TiO2 NPs were successfully embedded into the thermo-responsive PNIPAM/PAA microgel. Either Ag-TiO2 NPs or the composite materials exhibited a high photocatalytic degradation efficiency on the pollutant Rhodamine B and significant anticancer potential under visible light irradiation.
Since the authors understood the comment of the reviewer about the lack of the title to precisely describe the content of the study, thus the title was redefined.
All the modifications have been highlighted in yellow in the revised version of the manuscript. The reference and the figure numbering have also been changed.
Point 1: In order to much increase understandability of this work, The used materials and fabrication methods have to be clearly presented. I suggest addition of the first figure to explain the used materials and their fabrication methods.
Response 1: The authors would like to thank the reviewer for this valuable and constructive comment. They enriched Section «2.1.3. Composite nanoparticles» with additional information. Also, in order to clarify the steps of the synthesis procedure, the authors prepared three additional Figures (Figure 1-3) that schematically represent the whole process.
Point 2: Superiority and/or specific features of the used materials in bio-related properties have to be discussed upon comparisons over the related materials reported in the past literatures. Such discussions would originality and novelty of the work much clearer.
Response 2: The authors would gratefully thank the reviewer for this remark. They added a related part (at the end of 3.3.1. Effect on cell proliferation), focusing on the superiority and the specific features of the selected materials.
Point 3: Addition of conceptual term to the title would increase innovative impression to this manuscript. I may suggest use of an emerging conceptual term, nanoarchitectonics, in the title (as post-nanotechnology concept, see https://pubs.rsc.org/en/content/articlelanding/2021/nh/d0nh00680g). For example, the title like ... Composite nanoarchitectonics of thermo-responsive titania with photo-induced anticancer properties ... may sound more innovative.
Response 3: The authors appreciate this comment and changed the title to be sure that it accurately describe the content of their study. The new title is: “Composite nanoarchitectonics of photo-activated titania-based materials with anticancer properties”
Point 4: Some figures are not in good resolution. Please improve figure quality.
2
Response 4: The authors would like to thank the reviewer for this valuable suggestion. They improved the figure quality. They provided all the figures in resolution of 300 dpi.
Point 5: In Figure 2, vertical axis would be Transmittance (a.u.) (not Relative Intensity).
Response 5: The authors are thankful for the comment. They changed the axis title to Transmittance (a.u.) instead of Relative Intensity.
Point 6: Please possibly add error bars to plots in several graphs such as Figures 5, 8, and 9.
Response 6: The authors appreciate reviewers’ suggestion and they have added errors bars, obtained from 3 independent experiments in Figure 5, 8, and 9 that have renamed to 8, 12 and 13 in the revised manuscript.
Point 7: Figure 7 has bad color choice. Green on black background does not have good contrast. Please use lighter color on black backgrounds. Scale bars on these figure cannot be seen well. Please much improve these features.
Response 7: The authors are thankful for this comment. The green was the default color of the TEM device, and that was the reason that the authors initially submitted the figure in this form, but they achieved to replace green with yellow that possesses very good contrast. They also improved the scale bars in order to be clear.
Point 8: Figure 7f cannot be seen well. This graph can be represented as a separate figure.
Response 8: The authors would like to thank the reviewer for this valuable and constructive comment. EDS analysis Ag-TiO2/ microgel composite was finally presented as a separate figure (Figure 11) with a lighter background also.

Reviewer 3 Report

Good results

However, only in vitro evaluation is not enough to publish.

One comment.

Is it possible to obtain close results in 3D vitro models? Because the animal experiment was not performed, 3D models mimicking the animal condition should be indicated and discussed to evaluate the further effects in vivo. This paper is not included animal experiments, which is not enough.

To discuss this, the reviewer recommends that these references be quoted for anticancer assessment and that the sentences are added. An additional experiment is not needed when the sentences and discussion are added.

Cancers 202012(10), 2754

https://doi.org/10.1016/j.biomaterials.2019.119744

Author Response

1
Response to Reviewer 3 Comments
The authors would like to thank the reviewer for the valuable comments and for giving them the opportunity to revise their manuscript.
They, also, would like to underline that the motivation of our research is to synthesize titania-based composite materials with anticancer potential under visible-light irradiation Particularly, titanium dioxide (TiO2) nanoparticles (NPs) chemically modified with silver, were embedded in a stimuli-responsive microgel (a cross-linked interpenetrating network (IP) network, that was synthesized by poly (N-Isopropylacrylamide) and linear chains of polyacrylic acid sodium salt), forming composite particles. The ultimate goal of this research and our future perspective is to develop a drug delivery system, based on optical fibers which could photoactivate efficiently NPs, targeting on cancer cells. The produced Ag-TiO2 NPs, the microgel and the composite materials were fully characterized. Our results indicated that Ag-TiO2 NPs were successfully embedded into the thermo-responsive PNIPAM/PAA microgel. Either Ag-TiO2 NPs or the composite materials exhibited a high photocatalytic degradation efficiency on the pollutant Rhodamine B and significant anticancer potential under visible light irradiation.
Since the authors understood the comment of the reviewer about the lack of the title to precisely describe the content of the study, thus the title was redefined.
All the modifications have been highlighted in yellow in the revised version of the manuscript. The reference and the figure numbering have also been changed.
Point 1: Good results. However, only in vitro evaluation is not enough to publish. One comment.
Is it possible to obtain close results in 3D vitro models? Because the animal experiment was not performed, 3D models mimicking the animal condition should be indicated and discussed to evaluate the further effects in vivo. This paper is not included animal experiments, which is not enough.
To discuss this, the reviewer recommends that these references be quoted for anticancer assessment and that the sentences are added. An additional experiment is not needed when the sentences and discussion are added.
Cancers 2020, 12(10), 2754
https://doi.org/10.1016/j.biomaterials.2019.119744
Response 1: The authors would like to thank the reviewer for this valuable and constructive comment. They appreciate the reviewer’ s suggestion and added a specific paragraph at the end of the part “3.3.2. Effect on cytotoxicity”, mentioning that among their future perspectives is to test the produced materials in a 3D system.

Round 2

Reviewer 1 Report

The authors have significantly improved the manuscript and addressed the comments raised. Recommended for publication 

Reviewer 2 Report

Replies and revisions are fine. The revised version becomes acceptable.

Reviewer 3 Report

Good revision.